# Left-right asymmetric and smaller right habenula volume in major depressive disorder on high-resolution 7-T magnetic resonance imaging

Seo-Eun Cho[1]☯, Chan-A Park[2]☯, Kyoung-Sae Na[1], ChiHye Chung[3], Hyo-Jin Ma[1], Chang-Ki Kang[4]‡*, Seung-Gul Kang[1]‡*

1 Department of Psychiatry, Gil Medical Center, Gachon University College of Medicine, Incheon, Republic of Korea, 2 Biomedical Engineering Research Center, Gachon University, Incheon, Republic of Korea, 3 Department of Biological Sciences, Konkuk University, Seoul, Republic of Korea, 4 Department of Radiological Science, College of Health Science, Gachon University, Incheon, Republic of Korea

☯ These authors contributed equally to this work.
‡ CKK and SGK are contributed equally to this work as corresponding authors.
* kangsg@gachon.ac.kr (SGK); ckkang@gachon.ac.kr (CKK)

**Data Availability Statement:** We fully disclosed our data at the following URL: https://www.kaggle.com/seoeuncho/habenula-volume-in-mdd.

## Abstract

The habenula (Hb) has been hypothesized to play an essential role in major depressive disorder (MDD) as it is considered to be an important node between fronto-limbic areas and midbrain monoaminergic structures based on animal studies. In this study, we aimed to investigate the differences in volume and T1 value of the Hb between patients with MDD and healthy control (HC) subjects. Analysis for the Hb volumes was performed using high-resolution 7-T magnetic resonance (MR) image data from 33 MDD patients and 36 healthy subjects. Two researchers blinded to the clinical data manually delineated the habenular nuclei and Hb volume, and T1 values were calculated based on overlapping voxels. We compared the Hb volume and T1 value between the MDD and HC groups and compared the volume and T1 values between the left and right Hbs in each group. Compared to HC subjects, MDD patients had a smaller right Hb volume; however, there was no significant volume difference in the left Hb between groups. In the MDD group, the right Hb was smaller in volume and lower in T1 value than the left Hb. The present findings suggest a smaller right Hb volume and left-right asymmetry of Hb volume in MDD. Future high-resolution 7-T MR imaging studies with larger sample sizes will be needed to derive a more definitive conclusion.

## Introduction

Major depressive disorder (MDD) is a common mood disorder characterized by the presence of a depressed mood or loss of interest lasting for more than 2 weeks [1]. MDD is the leading cause of disability contributing to the overall burden of disease globally and causes emotional distress, functional impairment, health problems, and suicide. The pathophysiology of MDD

**Funding:** This research was supported by a grant from the Korea Health Technology R&D Project through the Korea Health Industry Development Institute (KHIDI), funded by the Ministry of Health & Welfare, Republic of Korea (grant number: HI17C2665). This work was also supported by a National Research Foundation of Korea (NRF) grant funded by the Korean government (MSIT), grant number NRF-2020R1A2C1007527. The funders had no role in study design, data collection and analysis, decision to publish, or preparation of the manuscript.

**Competing interests:** The authors have declared that no competing interests exist.

has been investigated with respect to neurotransmitter abnormalities, genetic causes, and structural and functional abnormalities in the brain. Recently, with the development of brain imaging, many studies of structural or functional abnormalities of the brain have been performed [2]. The reduction of brain volume in the hippocampus, orbitofrontal cortex, and subgenual prefrontal cortex has been consistently reported in previous studies [2–6].

The habenula (Hb) is thought to be involved in the etiology of MDD based on recent animal studies [7, 8]. It is an epithalamic structure and anatomical hub located at the center of the dorsal diencephalic conduction system, a pathway linking the forebrain to midbrain regions [9]. The Hb is considered to be related to punishment and motor activity during reward processes and the adaptive response to stress [10, 11], and based on previous experiments with animals, it is hypothesized to play an important role in controlling emotions; notably, the over-activation of the Hb is reported to be associated with depression [12–14]. Consistent with the monoamine hypothesis of MDD, the Hb is thought to regulate the activity of ascending mono-aminergic projections from the brainstem [15], and increased metabolism of the Hb has been observed in animal models of stress and depression [16]. In animal models of depressive disorder, lesions of the Hb induced decreased depression-like behavior and increased serotonin concentrations in the dorsal raphe [17]. The left and right Hbs are connected by the habenular commissure and can be functionally distinguished by the lateral and medial Hb [18]. The Hb receives inputs from the limbic system and basal ganglia mainly through the stria medullaris, and projects to midbrain regions through the fasciculus retroflexus [19].

However, the human Hb, which is estimated to be an important brain region involved in depression, is 5–9 mm in size and neuroimaging studies are lacking because the visualization and exact delineation of this structure is not easy using conventional 3-T brain magnetic resonance imaging (MRI). A postmortem brain study revealed a smaller neuronal cell area and number in the right Hb in patients depression ($n = 14$) than in healthy controls (HCs, $n = 13$) and patients with schizophrenia ($n = 17$) [20]. Patients with depression showed smaller right medial ($-24\%$) and lateral ($-20\%$) Hb volumes than those in HCs and patients with schizophrenia [20]. Recently, several studies on the structure of the Hb in depression using brain imaging have been attempted but have produced inconsistent results [21–23]. Unmedicated patients with bipolar depression showed significantly smaller Hb volumes than an HC group, and currently depressed females with MDD had smaller Hb volumes than healthy females [23]. In another study, hemispheric Hb volumes did not differ between medicated and unmedicated MDD patients and HCs; however, there were significant positive correlations between Hb volumes and depression severity [22]. A recent analysis showed that women with first-episode MDD had higher overall Hb white matter volumes than did healthy controls and patients with treatment-resistant/chronic MDD [21]. Perhaps the inconsistency might be due to limitations in the studies, such as a small number of subjects and insufficient resolution of the imaging modality. Therefore, the Hb needs to be researched using high-resolution MRI and precisely segmented.

Asymmetrical structural circuits or functional laterality of the Hb have been identified in vertebrates through numerous studies and experiments. Various species exhibit left-right asymmetry in the differentiation of habenular neurons, especially fish and amphibians; however, in humans, some studies have shown that both Hbs are symmetrically small and poorly developed [24] and in another study, the left lateral Hb was larger than the right by approximately 5% [25]. The possibility of asymmetry in functional connectivity of the human Hb was reported in a study performed using cardiac-gated resting state imaging [26]. In addition, a new research topic with clinical significance besides asymmetry in the Hb is myelin and T1 relaxation time, which is a measure of how quickly the net magnetization vector recovers to its ground state. After the radiofrequency (RF) pulse is stopped, the absorbed energy is released to

the surrounding tissue, and the protons are rearranged in the direction of the external magnetic field in equilibrium [27, 28]. T1 relaxation time is known to be sensitive to the water content in tissues and the macromolecular tissue environment. Changes in water composition can lead to obvious changes in the T1 relaxation time, and measurement of the T1 relaxation time provides a quantitative indicator of the physiological state of the underlying tissue [29, 30]. These T1 relaxation constants have been widely used to study high-intensity lesions such as multiple sclerosis and are used as markers of tissue characteristics in MR imaging studies [31]. A previous study using the T1 relaxation time was also conducted in patients with depression, and unipolar depression patients showed a decreased T1 in the hippocampus compared to healthy controls [32]. In that study, the researchers suggested that the biophysical tissue change might be due to depression [32]. In addition, quantitative T1 mapping has shown the possibility of elucidating cortical myelin content [33], especially in high-resolution 7-T MRI [34]. Strotmann et al. concluded that the low T1 value of the Hb was due to the unusually large number of myelinated fibers [35]. It is likely that decreases in R1 (R1 = 1 / T1 value) are attributable to decreases in myelin in depression [36]. In this context, investigators have begun to use T1 values and R1 as an in vivo assay of myelin content in the Hb [36, 37].

Therefore, the aims of our study were to investigate (i) whether Hb volume differs between patients with MDD and HCs and (ii) whether there is left-right asymmetry in Hb volume in each group. Our hypothesis is that, like the postmortem model, the right Hb volume is smaller in the MDD group and there is left-right asymmetry.

## Materials and methods

### Subjects

Patients with MDD and HCs ranging from 20 to 65 years in age participated in this study after providing informed written consent before the experiment. All participants were recruited from the psychiatric department of Gil Medical Center, Incheon, South Korea, via physician-initiated requests or referrals, or an in-hospital and online advertisement. All participants were initially screened using telephone interviews by a well-trained researcher. After the initial pre-screening, board-certified psychiatrist (SGK) interviewed all participants to assess their eligibility for the present study using the Structured Clinical Interview for Diagnostic and Statistical Manual of Mental Disorders, 5th edition (SCID) [38]. In addition, after the participants completed a self-report patient questionnaire for the SCID-5-personality disorder (SCID-5-SPQ) [39], the clinician (SGK) reviewed it and asked additional questions to exclude any participants with personality disorder. This study was approved by the Intuitional Review Board (IRB No. GDIRB2018-005) of the Gil Medical Center.

The common eligibility criteria are follows: (i) aged 20–65 years; (ii) identified as right-handed using the Edinburgh Handedness Test [40]; (iii) no unstable or major medical or neurological disorders within the past 1 year; (iv) no history of cerebrovascular accident; (v) no history ono substance use disorder within the past 1 year; (vi) no personality disorder, intellectual disability, or neurocognitive disorders; (vii) no current serious suicide risk; (viii) no history of significant brain injury or previous abnormal findings on brain imaging; (ix) no relative or absolute contraindications for MRI (e.g., metal material in the body); and (x) not pregnant or lactating.

The MDD patients who met the diagnostic criteria for MDD as stated in the DSM-5 [1] were included. Patients who had hypomanic or manic episodes were excluded. In addition, participants in the MDD group did not have any of the following psychiatric comorbidities: schizophrenia spectrum and other psychotic disorders (delusional disorder, brief psychotic disorder, schizophreniform disorder, schizophrenia, schizoaffective disorder, catatonia);

major anxiety disorders (panic disorder, social anxiety disorder, specific phobia); obsessive-compulsive and related disorders (obsessive-compulsive disorder, body dysmorphic disorder, hoarding disorder, trichotillomania, excoriation disorder); or disruptive, impulse-control, and conduct disorders (oppositional defiant disorder, intermittent explosive disorder, conduct disorder). If a patient's psychiatric symptoms could be explained as being due to MDD, we did not specify an additional diagnosis for the patient according to DSM-5.

The MDD group was divided into four subgroups as follows: (i) first episode (those who have experienced the first episode of MDD and have not previously taken psychotropic drugs such as antidepressants, and have a 17-item Hamilton Depression Rating Scale (HDRS-17) score of 7 or higher); (ii) treatment-resistance (those who have experienced major depressive episodes for more than two years in their lifetime and who have sustained major depressive episodes that do not show treatment response to two or more antidepressants); (iii) recurrence (those with two or more episodes of major depressive episodes and have not taken psychiatric medications such as antidepressants for more than one month); and (iv) remission (those with an HDRS-17 score ≤ 6 over the last 2 months or more).

The HCs were included according to the following criteria: (i) no symptoms or history of psychiatric disorders; (ii) a total score ≤ 6 on the Hamilton depression rating scale (HDRS-17) at screening; (iii) no history of taking psychotropic medications during their lifetime; and (iv) no first-degree relatives with a major psychiatric disorder such as a mood disorder or schizophrenia. The MDD and HC groups were matched for age and sex.

For all participants, depressive symptom severity was measured using the validated version of the HDRS-17 [41], Beck depression inventory (BDI) [42], and clinical global impression scale (CGI) [43, 44]. As per a previous study [41], severity on the Korean version of the HDRS-17 was defined based on the total score as follows: within the normal range (0–6), mild depression (7–17), moderate depression (18–24), and severe depression (≥ 25).

## Image acquisition

Whole-brain sagittal images were acquired using an 8-channel phased-array coil for 7-T MRI (MAGNETOM 7T, Siemens, Erlangen, Germany). The subjects were asked not to move their heads while they were kept in a comfortable lying position to minimize movement during the scan. In addition, cushions were placed between the RF coil and the subject's head to secure the head. In this study, to evaluate the possibility of simultaneously recording relaxation times, such as T1 and T2*, we utilized the prototype multi-echo (ME) magnetization-prepared 2 rapid gradient echoes (MP2RAGE) sequence provided by Siemens [45]. Although magnetization-prepared rapid gradient echo (MRPAGE) is commonly used for anatomical imaging, MP2RAGE, a new imaging technique that has the advantage of obtaining T1 relaxation time maps, was used. However, it was not expected to distinguish small and low-contrast structures such as the Hb. Therefore, it was necessary to acquire both images using MPRAGE and MP2RAGE imaging techniques for comparison. The acquisition was performed using the following parameters: field of view (FOV) = $166 \times 166 \times 135.2$ mm$^3$ with nominal isotropic resolutions of 0.65 mm; matrix size = $256 \times 256$; 208 slices along the right-left axis (sagittal orientation); repetition time (TR) = 8000 ms; two inversion times (TIs) = 1000/3200 ms; flip angle (FA) = 4˚; four echo times (TEs) = 3.46, 7.28, 11.1, and 14.92 ms; bandwidth = 280 Hz/px yielding an acquisition time (TA) = 14 min 16 s; bipolar readout; generalized auto-calibrating partially parallel acquisitions (GRAPPA) with accelerating factor = 3 (50 reference lines); and 7/8 and 6/8 partial Fourier factors along the phase-encoding (PE) and slice-encoding (SE) directions, respectively. To obtain reference anatomical data, a further scan was made in the same session using conventional 3D T1 magnetization-prepared rapid gradient echo

(MPRAGE) using the same orientation (sagittal), number of excitations (NEX = 1), GRAPPA acceleration, and partial Fourier along the SE direction.

## Manual delineation of the habenula on 7-T T1 map and calculation of T1 values

Two well-trained researchers performed the manual delineation of the right and left Hbs using the 7-T MR images (i.e., T1 map) from the subjects. At first, the left and the right Hb were identified based on the selected slices in each data. They were manually traced from the T1 map image according to the following procedure. Using MRIcron and ImageJ as analytic tools, the examiners were able to see the image in all three planes (sagittal, coronal, and axial) simultaneously and to segment manually the target voxels outlining the Hb surface where the signal intensity differs clearly from that of the adjacent brain tissues. These data were used to evaluate the reliability of the Hb definition. During the trace, extra care was taken to separate the Hb boundaries from adjacent, non-Hb brain tissues, such as white matter and cerebrospinal fluid (CSF). This was done as precisely as possible, as confirmed by other senior researchers who had the sufficient research experience in brain MRI. The reliability was determined using the overlap index ratio (%) suggested by a previous study [46]. The volumes of the segmented structures were utilized for inter-rater reliability (interclass correlation [ICC]), and the overlapping ratio of the manually segmented whole Hb was calculated to ensure that it was more than 70%. Then, the area was confined to the overlapped areas, so as not to overestimate the habenular volume [46].

$$\text{Overlap Index Ratio } (\%) = (A \cap B)/(A \cup B) \times 100$$

where A is the number of voxels selected by Examiner #1 and B is the number selected by Examiner #2. The number of voxels (#) within overlapping areas was counted, and then the volume was calculated with the following formula: # × voxel size, where is 0.65×0.65×0.65 (imaging resolution). MRIron (https://www.nitrc.org/projects/mricron) and MATLAB (The Mathworks, Inc.) software were used for segmentation and volume calculations, respectively.

In the data obtained from MPRAGE and ME-MP2RAGE T1 map images, the signal intensity and contrast were different according to the different MR sequences. As shown in the magnified images of **Fig 1**, the CSF is seen as the darkest signal in the MPRAGE image, but the brightest in the ME-MP2RAGE T1 map image. In addition, compared to the surrounding tissues, the Hb has brighter signals in the MPRAGE image and darker signals in the ME-MP2-RAGE T1 map image. Overall, the contrast of the ME-MP2RAGE T1 map image was greater than that of the MPRAGE image (**Fig 1**).

The MPRAGE ($0.5 \times 0.5 \times 0.5$ mm$^3$) resolution was better than that of the ME-MP2-RAGE T1 map ($0.65$ x $0.65$ x $0.65$ mm$^3$), but the contrast-to-noise ratio (CNR) of the ME-MP2RAGE T1 map was relatively higher to better distinguish the tissue surrounding the Hb on the T1 map. Therefore, the two examiners (Examiner #1 and Examiner #2) manually performed Hb segmentation of the ME-MP2RAGE T1 map images. The Hb segmentation results of the two examiners are shown in each representative sample in both (a) HC and (b) MDD groups in **Fig 2.** Although there might be variations between examiners in the voxels at the edge of the Hb, the overlap voxels (green) were localized mainly in the center. Each voxel has a T1 value estimated from the acquired MRI images, which was introduced in previous studies [45, 47]. The final T1 value for each subject was the average of the T1 values within the overlapped area.

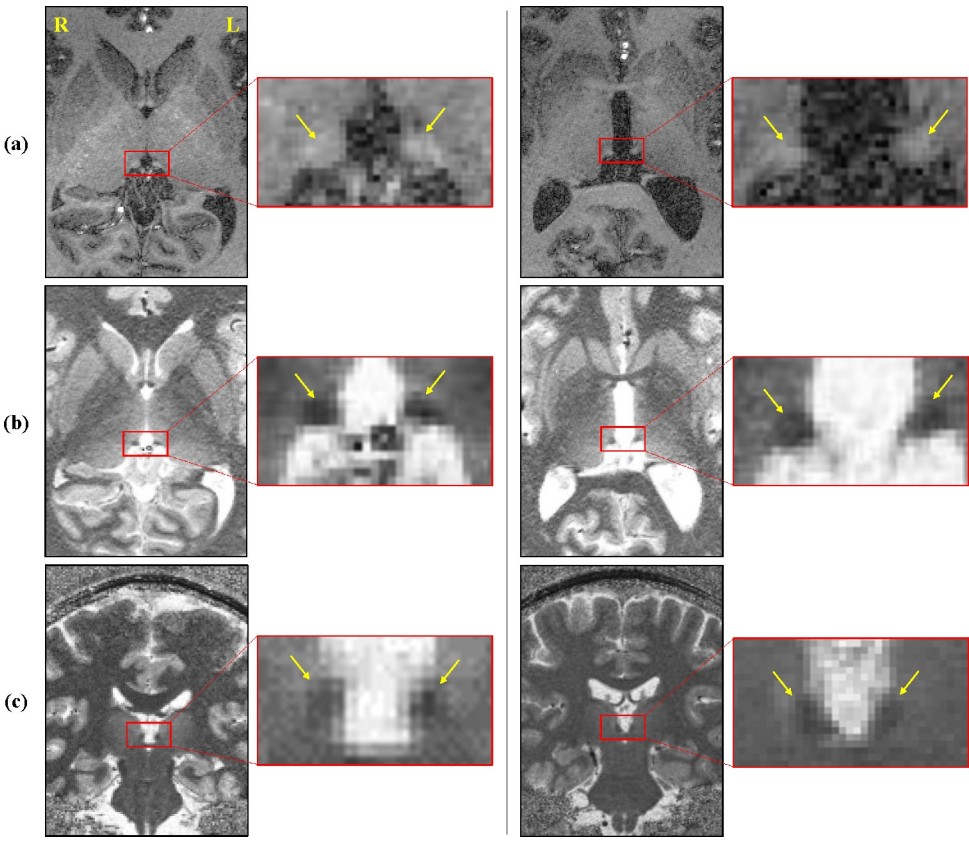

**Fig 1. The habenula of an HC (left panel) and MDD patient (right panel).** (a) axial MPRAGE, (b) axial view in ME-MP2RAGE T1 maps, and (c) coronal view in ME-MP2RAGE T1 maps. Compared to the surrounding tissues, the habenula has a brighter signal on the MPRAGE image and darker signal on the ME-MP2RAGE T1 map image. Overall, the contrast on the ME-MP2RAGE T1 map image is greater than that on the MPRAGE image. The yellow arrow points to the habenula, and the boundaries of the habenula are indicated by the yellow dotted line. Abbreviations: HC, healthy control; MDD, major depressive disorder; ME-MP2RAGE, multi-echo magnetization-prepared 2 rapid gradient echoes; MPRAGE, magnetization-prepared rapid gradient echo.

## Statistics

Demographic data and clinical characteristics, including age, sex, education level, and scores of the clinician-administered scales of the HDRS-17, BHS, BDI, CGI, and SSI were summarized and compared using a Student's t-test or chi-square test. In the MDD group, the number of depressive episodes, duration of current depressive episode, duration of illness, whether taking antidepressants, the duration of antidepressant use, and the subgroups (first episode, treatment-resistance, recurrence, and remission) of MDD were described.

We estimated the volume (mm$^3$) and T1 value of the right and left Hbs from the overlapping voxels of the two examiners. To adjust for potential confounding factors, analysis of covariance (ANCOVA) was conducted comparing the normalized right and left Hb volumes between the MDD and HC groups, controlling for age, sex, education level, and individual differences in brain size. The normalization of the Hb volumes was performed using the total intracranial volume (ICV) from the 3T MRI. The Hb volumes were divided by the ICV for each participant ($\frac{Hb\ Volume}{ICV} \times 100$) to adjust for individual differences in brain size. Since the number of regions we investigated in this analysis was two, the Bonferroni adjusted level of significance (alpha) was calculated to be 2.5%. We rejected the null hypothesis when the p-value was less than the adjusted alpha.

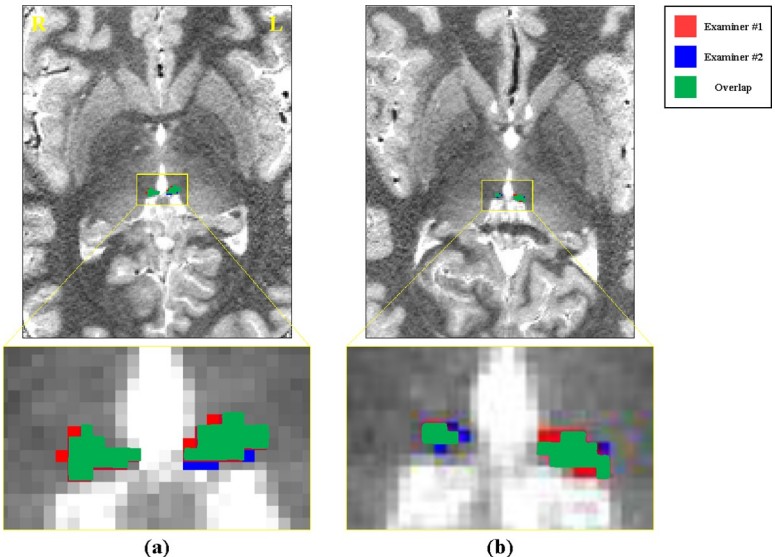

**Fig 2.** The right and left overlapping habenula in an (a) HC and (b) MDD patient on an axial ME-MP2RAGE T1 map segmented by Examiner #1 and Examiner #2. Red, blue, and green indicate the voxels segmented by Examiner #1, Examiner #2, and the overlap of voxels identified by both examiners, respectively. Abbreviations: R, right; L, left; HC, healthy control; MDD, major depressive disorder; ME-MP2RAGE, multi-echo magnetization-prepared 2 rapid gradient echoes.

Furthermore, the group difference (MDD vs. HC for volume and T1 value of the right and left Hbs) and the inter-hemispheric difference (right vs. left Hb volume and T1 value) in each group were analyzed using Student's t-test and paired t-test, respectively. Normality was confirmed using the Kolmogorov-Smirnov method. When not satisfying the homogeneity of variance using Levene's test, the difference was tested using Welch's t-test.

In addition, we performed multiple linear regression analysis to determine whether demographic and clinical characteristics affect the Hb volume of the MDD group. In multivariate regression analyses, variables such as age, sex, years of education, the use of antidepressants, duration of antidepressant use, duration of illness, illness recurrence, HDRS-17 score, SSI score, and depression severity were set as independent variables with the stepwise method as they were significantly correlated with whole Hb volume. Whole Hb volume was used as a dependent variable in multivariate linear regression analyses.

The Statistical Package for the Social Sciences (SPSS, IBM Inc.) program was used and $P < 0.05$ was set as the limit for statistical significance.

## Results

The demographic and clinical characteristics and comparison of the groups of subjects with MDD (n = 33) and HCs (n = 36) are shown in **Table 1**. There was no significant difference in the sex ratio or age between the two groups (**Table 1**). The years of education was significantly lower in the depressed group than in the control group ($P = 0.001$, see **Table 1**). The MDD patients had significantly more severe depressive symptoms as measured using the HDRS and BDI ($P < 0.001$, see **Table 1**). In addition, the BHS, CGI, and SSI scores were also higher in patients with MDD than in HCs ($P < 0.001$, see **Table 1**). Compared to the control group, the depressed group also reported more hopelessness and suicidal ideation, and the overall clinical impression was more severe. When the severity of depression was classified according to the range of the total score on the HDRS-17, 38 (55%), 14 (20%), 13 (19%), and 4 (6%) were classified as mild, moderate, and severe depression, respectively.

**Table 1. Demographic and clinical characteristics of the MDD and HC groups.**

| Clinical variables | MDD (n = 33) | HC (n = 36) | P values |
|---|---|---|---|
| Age at scan, years (mean ± SD) | 40.58 ± 14.16 | 35.17 ± 12.00 | 0.093[a] |
| Sex (male: female) | 8: 25 | 12: 24 | 0.406[b] |
| Education, years (mean ± SD) | 12.73 ± 3.45 | 15.22 ± 1.91 | 0.001[a] |
| Severity ranges for the HDRS-17 score, n (%) | | | |
| Within the normal range (0–6) | 2 (6) | 36 (100) | N/A |
| Mild depression (7–17) | 14 (42) | 0 | N/A |
| Moderate depression (18–24) | 13 (39) | 0 | N/A |
| Severe depression (≥ 25) | 4 (12) | 0 | N/A |
| The number of episodes (mean ± SD) | 2.58 ± 1.39 | N/A | N/A |
| Duration of current depressive episode, weeks (mean ± SD) | 73.93 ± 16.12 | N/A | N/A |
| Duration of illness, months (mean ± SD) | 63.45 ± 57.45 | N/A | N/A |
| The use of psychotropic drugs, n (%) | | | |
| Antidepressants | 27 (82) | N/A | N/A |
| Benzodiazepine | 19 (58) | N/A | N/A |
| Second-generation antipsychotics | 14 (42) | N/A | N/A |
| Duration of antidepressant use, weeks (mean ± SD) | 82.18 ± 26.24 | N/A | N/A |
| MDD subgroups[c], n (%) | | | |
| First episode | 12 (36) | N/A | N/A |
| Treatment-resistance | 6 (18) | N/A | N/A |
| Recurrence | 15 (45) | N/A | N/A |
| Remission | 2 (6) | N/A | N/A |
| HDRS-17 score (mean ± SD) | 15.95 ± 5.40 | 2.53 ± 2.41 | < 0.001[a] |
| BHS score (mean ± SD) | 12.52 ± 5.93 | 1.19 ± 1.47 | < 0.001[a] |
| BDI score (mean ± SD) | 27.79 ± 12.66 | 3.50 ± 3.68 | < 0.001[a] |
| CGI score (mean ± SD) | 4.03 ± 0.95 | 1.03 ± 0.17 | < 0.001[a] |
| SSI score (mean ± SD) | 14.00 ± 8.69 | 1.00 ± 1.53 | < 0.001[a] |

[a] Student's t-test

[b] Chi-square test

[c] Some patients belonged to more than one group.

Data are presented as means ± standard deviations or numbers (percentages) unless otherwise indicated.

Abbreviations: BAI, Beck Anxiety Inventory; BDI, Beck depression inventory; BHS, Beck Hopelessness Scale; CGI, clinical global impression scale; HDRS, Hamilton Depression Rating Scale; HC, Healthy Control; MDD, Major Depressive Disorder; SD, Standard Deviation; SSI, Scale for Suicide Ideation.

In the MDD group, the average duration of the current episode was 73.93 weeks, the average number of depressive episodes was 2.58 (SD 1.39), the proportion of treatment-resistant patients was 15%, and the average disease duration was 63.45 months. The MDD group was divided into four subgroups: first episode (n = 12), treatment-resistance (n = 6), recurrence (n = 15), and remission (n = 2). Since some patients belonged to more than one group, the sum of the number of participants of each group exceeded the total number of patients. Eighty-two percent of the MDD group were taking antidepressants, and the average duration of antidepressants use was 82.18 weeks (Table 1). The main antidepressants in patients with MDD were escitalopram (n = 12), vortioxetine (n = 3), bupropion (n = 2), desvenlafaxine (n = 2), fluoxetine (n = 2), sertraline (n = 2), milnacipran (n = 2), and mirtazapine (n = 2). In the MDD group, other psychotropic drugs were being taken as follows: benzodiazepine (n = 17), zolpidem (n = 2), quetiapine (n = 7), aripiprazole (n = 6), and olanzapine (n = 1). Second-generation antipsychotics were prescribed as adjunctive treatments for MDD.

**Table 2. Comparison of the normalized habenula volume between the MDD and HC groups, controlling for potential confounders.**

| Volume Ratio | MDD (n = 33) | | HC (n = 36) | | F | P | partial η² |
|---|---|---|---|---|---|---|---|
| | M | SD | M | SD | | | |
| Right Hb | $12.43 \times 10^{-4}$ | $2.28 \times 10^{-4}$ | $12.88 \times 10^{-4}$ | $2.79 \times 10^{-4}$ | 5.544 | 0.022* | 0.080 |
| Left Hb | $12.75 \times 10^{-4}$ | $2.26 \times 10^{-4}$ | $12.87 \times 10^{-4}$ | $3.52 \times 10^{-4}$ | 0.037 | 0.848 | 0.001 |

Statistical analysis was performed using an analysis of covariance. Analyses were adjusted for age, sex, and education level. The normalization of the Hb volumes was performed by dividing the habenula volumes by the ICV for each participant ($\frac{Hb\ Volume}{ICV} \times 100$) to adjust for individual differences in brain size.

* indicates a significant difference ($P < 0.025$).

Abbreviations: HC, healthy control; MDD, major depressive disorder; M, mean; SD, standard deviation; partial η², partial eta-squared; ICV, intracranial volume; Hb, habenula.

Hb volume and T1 values were calculated from overlapping voxels in the right and left Hb regions across all subjects. The mean overlapping volumes in each Hb are 15.63 to 19.98 as shown in **Tables 2 and 3, Fig 3**, respectively. The ICC coefficients were 0.703 ($P < 0.001$) and 0.656 ($P < 0.001$) in the right and left Hb, respectively. The average overlap index ratio of the quantitative analysis by the two examiners are as follows: the overlap index ratio (mean ± standard deviation [SD]) of the two examiners was 71.85 ± 7.47 in the right Hb and 73.76 ± 7.96 in the left Hb (MDD group; n = 33) and 70.90 ± 8.22 in the right Hb and 69.92 ± 6.87 in the left Hb (HC group; n = 36). In **Fig 3**, the volume and T1 values of the right and left Hb in each group are visualized as scatter plots including the mean and SD.

In the between group comparison, according to the results of the ANCOVA, which included age, sex, and education level as covariates, there was a significant difference in the right Hb volume ratio between the MDD and HC groups (F = 5.544, $p = 0.022$, **Table 2**). The partial eta-squared values (representing the effect size) for the left and right Hb volume ratios were small ($\eta^2 = 0.001$) and relatively large ($\eta^2 = 0.080$), respectively. In the analysis using Student's t-test, participants with MDD had a significantly smaller right Hb volume than those in the HC group ($P = 0.018$, **S1 Table**). There was no significant difference in the left Hb volume between the two groups. This is consistent with the results from the ANCOVA, in which the volume of the right Hb differed significantly between the two groups.

The inter-hemispheric difference in Hb volume and T1 value between the left and right Hbs was compared in each group. There was a significant difference in the Hb volume and T1 value between the left and right Hbs only in the MDD group ($P = 0.010$ in volume and $P < 0.001$ in T1 value, **Table 3**). In the HC group, there was a significant difference in T1 value between the left and right Hbs ($P = 0.033$); however, there was no significant inter-hemispheric volume difference ($P = 0.928$, **Table 3**).

**Table 3. Comparison of the volume and T1 value between right and left habenula segmented by both examiners between each group.**

| | Group | Right | Left | t score | P values |
|---|---|---|---|---|---|
| Volume (in mm³ ± SD) | MDD (n = 33) | 16.43 ± 3.24 | 18.37 ± 3.39 | −2.722 | 0.010* |
| | HC (n = 36) | 18.54 ± 3.91 | 18.47 ± 5.04 | 0.092 | 0.928 |
| T1 (value ± SD) | MDD (n = 33) | 1200.96 ± 41.29 | 1225.03 ± 34.81 | −5.503 | < 0.001* |
| | HC (n = 36) | 1207.84 ± 32.35 | 1220.51 ± 31.90 | −2.225 | 0.033* |

* indicates significant difference ($P < 0.05$).

The statistical analysis was performed using a paired t-test.

Abbreviations: HC, healthy control; MDD, major depressive disorder; SD, standard deviation.

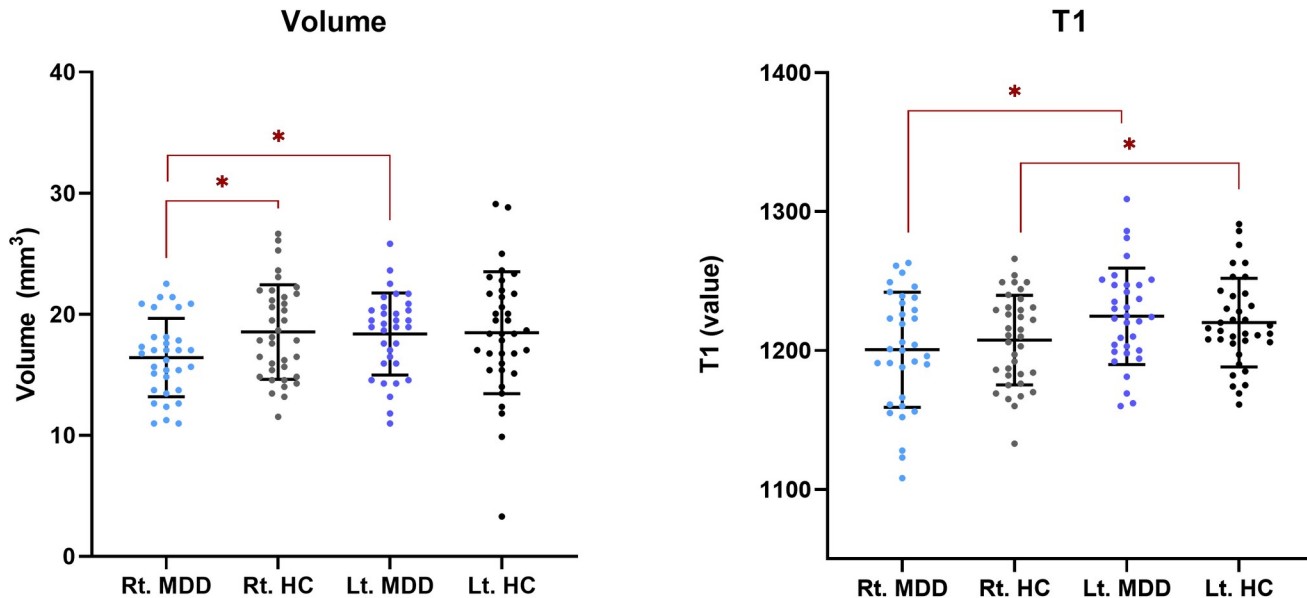

**Fig 3. Scatter plot with the mean and standard deviation of the volume and T1 value of the right and left habenula in each group.** Volumes or T1 value of the hemispheric habenula (Y axis) for the diagnostic groups (X axis). The graph plots the individual data points with a horizontal line superimposed at the mean and error bars showing plus and minus one SD of each group. We drew red lines between the combinations that differed significantly and marked them with an asterisk (*). Please note the smaller right habenula volume in the MDD group than the left habenula volume in the MDD group and those of the right and left habenula in HCs. Also, note that the T1 value in both the HC and MDD groups is lower in the right habenula than in the left. Abbreviations: Rt, right; Lt, left; HC, healthy control; MDD, major depressive disorder; SD, standard deviation.

The multivariate linear regression analysis showed the final model was suitable ($F = 12.147$, $P < 0.001$) with 62.3% explanatory power (adjusted $R^2 = 0.623$). In the MDD group, whole Hb volume was associated with age ($P = 0.011$), sex ($P = 0.001$), years of education ($P = 0.036$), and duration of antidepressant use ($P = 0.014$). Among these independent variables, it was found that sex ($\beta = -0.485$) had the relatively highest influence on Hb volume, followed by age ($\beta = -0.371$), duration of antidepressant use ($\beta = 0.331$), and years of education ($\beta = 0.299$). (Table 4).

**Table 4. Factors associated with whole habenula volume in the MDD group.**

| Independent variable | Unstandardized coefficients | | Standardized coefficient | *P* |
|---|---|---|---|---|
| | *B* | SE | β | |
| (Constant) | 1585876.702 | 132694.373 | | **< 0.001***\*\*\* |
| Age | −4578.510 | 1650.256 | −0.371 | **0.011**\* |
| Sex (female) (Reference group: male) | −195964.120 | 48694.135 | −0.485 | **0.001**\*\* |
| Years of education | 14299.465 | 6417.061 | 0.299 | **0.036**\* |
| Duration of antidepressant use | 401.717 | 150.203 | 0.331 | **0.014**\* |
| | $F = 12.147$, $P < 0.001$, adjusted $R^2 = 0.623$, DW = 2.105 | | | |

Final models of multivariate linear regression analyses with stepwise method of whole habenula volume.

*$P < .05$

**$P < .01$

***$P < .001$.

The dependent variable was habenula volume (log-transformed); independent variables were age, sex (female), years of education, the use of antidepressants (yes/no), duration of antidepressant use, duration of illness, recurrence of depression, HDRS-17 score, SSI score, and depression severity (mild/moderate/severe).

Abbreviations: DW, Durbin-Watson; HDRS, Hamilton Depression Rating Scale; SE, standard error; SSI, Scale for Suicide Ideation.

## Discussion

The main findings of this study are that MDD patients have a smaller right Hb than that of HCs; however, there is no significant difference in the left Hb between groups. In addition, in the MDD group, the volume of the right Hb was significantly smaller than that of the left Hb. In addition, in the analysis that included age, sex, and education level as covariates, the right Hb volume ratio was significantly different between the healthy and depressed groups. Thus, unlike that in the HC group, there was inter-hemispheric asymmetry in the volume of the Hb in the MDD group. Moreover, among the depressed group, it was found that the whole Hb volume was smaller for females and those who were older than for males and those who were younger, respectively, and that the Hb volume increased as the years of education or the duration of antidepressant use was longer.

After the report that the right Hb volume of postmortem brains from unipolar and bipolar depression patients was smaller than that in the healthy control group [20], in vivo human brain imaging studies of patients with MDD examined the difference in Hb volume, and three such studies were published. One study found no significant difference in the Hb volume between MDD patients and HCs; only unmedicated females with MDD had a smaller absolute Hb volume than female HCs on 3-T MRI [23]. The volumetric difference in that study was driven by a decrease in right Hb volume [23]. In a 3-T MRI study, women with first episode MDD had a larger Hb volume in the white matter than HCs and patients with treatment-resistant MDD [21]. The first 7-T MRI study hypothesized that there would be a lower right Hb volume in MDD patients; however, they made no such finding [22]. Instead, there was a significant positive correlation between bilateral Hb volume and disease severity scored using the HDRS and BDI in unmedicated MDD patients [22]. As such, the results of the three previous studies of Hb volume were inconsistent.

The present results are in agreement with the results—smaller volume of the right Hb—of the postmortem study by Ranft et al. [20] In rodent studies, repeated stressor-induced elevated adrenal steroid secretion, which increased N-methyl-D-aspartate (NMDA) receptor signaling, led to dendritic atrophy and neuronal cell death of the hippocampus, medial prefrontal cortex, and amygdala in a neurotoxic process [48]. In fact, Ranft et al. reported a significant reduction (up to 40%) in neuronal counts in the medial Hb [20]. In patients with MDD, with more frequent exposure to stress, the glial cells of the Hb may be damaged by excessive neuroinflammation, oxidative stress, and excitatory toxicity, leading to a decrease in neuropils and reduced brain volume [49]. Since patients with MDD are more susceptible to stressors, their amygdala and hypothalamic pituitary adrenal axis might be hyperactivated in response to stressful events. This can eventually increase cortisol secretion and suppress immune function [50].

As with previous studies by Savitz et al. [23], it was shown that depressed women had a smaller Hb volume. Another study reported that the Hb volume was smaller when the depression was recurrent or chronic in depressed women [21]. Compared to men in the MDD group, women are affected by female sex hormones, are more sensitive to socio-cultural stressors, and have limited methods of relieving stress [51], they might be more vulnerable to Hb volume reduction. In addition, the possibility of sex differences in Hb volume irrespective of depression cannot be excluded [52]. Unlike previous studies that did not find a correlation of Hb volume and age [21, 22], this study showed that the older the individuals in the depressed group, the smaller the volume. The older the patients with MDD, the more likely they are to live in isolation, have less interpersonal contact, and have a higher likelihood of physical illness [53], which may lead to a smaller whole Hb volume. To our knowledge, there are no research results that clearly demonstrate a correlation between Hb volume and duration of education or antidepressant medication use. Past studies suggest that use of antidepressants might have

an effect on preventing neuronal damage and cell loss that occur in patients who are depressed [54, 55]. In this context, the longer period of antidepressant use in patients with MDD might be related to a lower decrease in Hb volume. Since pubertal hormones are released and development of higher-order cognition through enhanced plasticity of cortical circuits is present in adolescents [56], the duration of education could affect the volume of the brain. Thus, the longer the education period, the greater the Hb volume.

The right Hb is smaller in volume than the left Hb in MDD and this result was also consistent with the results of the postmortem studies [20]. The selective reduction of the neuronal cell numbers of the right Hb could be attributed to an inter-hemispheric imbalance in depression. Depression is associated with a hyperactive right hemisphere and relatively hypoactive left hemisphere [57]. In a previous animal model study using rats, reduced serotonin connectivity of the dorsal raphe with the Hb that was predominantly pronounced in the right hemisphere was reported and this was interpreted as asymmetric hemispheric involvement in depression [58]. In humans, the association between treatment response to depression and connectivity between the Hb and other brain regions is thought to show left-right Hb asymmetry. The positron emission tomography study in treatment-resistant MDD showed decreased metabolism in the right Hb and the medial and orbital prefrontal networks to ketamine administration [59]. Compared to responders, treatment non-responders had lower fractional anisotropy in the right Hb afferent fibers and lower resting state functional connectivity (RSFC) between the right Hb and median raphe, but higher RSFC between the left Hb and locus coeruleus [60]. Laterality differences might be necessary to preserve good somatosensory processing in MDD [61] and various neurological conditions are associated with loss in the lateralization of brain activity [62–64]; however, it is unclear whether these abnormalities are a cause or consequence of the disease.

T1 relaxation time showed a left-right asymmetry in both the MDD and HC groups. T1 relaxation time is a measure of how quickly the longitudinal magnetization recovers to its ground state. Previous studies have reported changes in the T1 relaxation time of the hippocampus in patients with depression, and it might be considered a biophysical tissue change due to depression [65]; however, the clinical significance and interpretation of the T1 value is still unclear.

Previous studies, including meta-analyses, have investigated structural brain abnormalities in depression [66–68]; however, our study is distinct from previous studies. In most previous studies, the Hb were not designated as ROIs, and most studies used lower resolution brain MRIs from 3T, 1.5T, or 1.0T scanners [66, 67]. In addition, the clinical diagnoses of the participants were different (unipolar depression in our study vs. bipolar depression in a previous study [68]), or the diagnostic system or clinical scales [66, 67] used in the studies were dissimilar to ours. Moreover, there were differences in the treatment settings, brain imaging techniques, and methods of obtaining and analyzing the ROIs between the current and previous studies [66–68].

This study has several advantages over previous neuroimaging studies of the Hb. The T1 MP2RAGE map images, compared to the MPRAGE images previously used for Hb segmentation, show much better contrast for the Hb. Compared to the MPRAGE image previously used for Hb segmentation, the ME-MP2RAGE T1 map image shows a much better CNR of the Hb region than that in previous studies [45, 47]. Moreover, in contrast to previous studies that segmented the Hb by referring to neighboring anatomical landmarks [69], this study performed manual delineation based on high signal intensity differences between the Hb and adjacent brain tissues and analyzed overlapping areas visually delineated by two raters, which is a more commonly used research method in neuroimaging studies [46]. In the future, it will be necessary to evaluate and compare the consistency of measured Hb volumes according to the

research method used. Each Hb volume was within approximately 20 mm$^3$ (15.63 to 19.98) (Table 2), which is similar to previously reported Hb volumes [70, 71]. The overlap index ratio of the Hb delineation ranged from 71.30 to 75.79. When the overlap index ratio is 70% or more, the segmentation of the two examiners is considered to be reliable [46].

## Limitations

Although the results of this study are thought to contribute to clarifying the etiology of MDD patients in the future, it is not likely that brain imaging studies can be applied to biomarker for the diagnosis, treatment response, and recurrence of MDD yet [72]. Due to using self-reported measures, not structured interviews, to screen for personality disorders, there is a possibility of self-report bias. Therefore, we might not have accurately excluded participants with personality disorders. In addition, although we attempted to exclude participants with psychiatric disorders other than MDD under the DSM-5, it would have been difficult to entirely exclude individuals who might have also had these other disorders. Thus, there is a possibility that structural changes in the brain were caused by comorbid disorders. Owing to the relatively small number of subjects in this study, it is difficult to generalize our results. In addition, more advanced neuroimaging analyses and techniques such as machine learning methods should be attempted to further reveal evidence regarding the association between depression and the Hb [73]. Additionally, the severity of MDD symptoms and the type, amount, and duration of psychotropic medications being taken by each participant in the MDD group might have affected the outcomes. In future research, it is essential to compare subdivided brain regions according to the patients' severity of symptoms, whether or not they are taking any medications, and medication type. Further replication studies with more control over such variables and larger sample sizes are needed.

## Conclusion

This study showed decreased right Hb volume and left-right asymmetry of the Hb volume and T1 value in MDD. Our results, obtained using high-resolution 7-T MRI, could contribute to a better understanding of the neuroanatomical characteristics of Hb in MDD. In addition, the study of the association between Hb and the monoamine center in humans is considered an important field to explore the etiology and treatment mechanism of depression.

## Supporting information

**S1 Table. Group differences in the volume and T1 value of the habenula segmented by both examiners between MDD and HC.** * indicates significant difference ($P < 0.05$). The statistical analysis was performed using a Student's t-test. Abbreviations: HC, healthy control; MDD, major depressive disorder; SD, standard deviation.
(DOCX)

## Author Contributions

**Conceptualization:** ChiHye Chung, Seung-Gul Kang.

**Data curation:** Kyoung-Sae Na, Seung-Gul Kang.

**Formal analysis:** Chan-A Park, Chang-Ki Kang.

**Investigation:** Seo-Eun Cho, Chan-A Park, Chang-Ki Kang, Seung-Gul Kang.

**Methodology:** Chan-A Park, Chang-Ki Kang, Seung-Gul Kang.

**Project administration:** Seo-Eun Cho, Chang-Ki Kang, Seung-Gul Kang.

**Resources:** Seung-Gul Kang.

**Supervision:** Kyoung-Sae Na, ChiHye Chung.

**Validation:** Kyoung-Sae Na, Seung-Gul Kang.

**Visualization:** Seo-Eun Cho, Chan-A Park, Hyo-Jin Ma, Chang-Ki Kang.

**Writing – original draft:** Seo-Eun Cho, Chan-A Park, Chang-Ki Kang, Seung-Gul Kang.

**Writing – review & editing:** Chang-Ki Kang, Seung-Gul Kang.

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
