## [Decision Letter · Decision Letter 0]

22 Sep 2020

PONE-D-20-18962

Left-right asymmetric and smaller right habenula volume in major depressive disorder on high-resolution 7-T magnetic resonance imaging

PLOS ONE

Dear Dr. kang,

Thank you for submitting your manuscript to PLOS ONE. After careful consideration, we feel that it has merit but does not fully meet PLOS ONE’s publication criteria as it currently stands. Therefore, we invite you to submit a revised version of the manuscript that addresses the points raised during the review process.

Please submit your revised manuscript in the next 3 months. If you will need more time than this to complete your revisions, please reply to this message or contact the journal office at plosone@plos.org. Please include the following items when submitting your revised manuscript:

We look forward to receiving your revised manuscript.

Kind regards,

Danilo Arnone

Academic Editor

PLOS ONE

Journal Requirements:

Reviewers' comments:

Reviewer #1: In this manuscript, the authors examined the differences in volume and T1 value of the habenula (Hb) between 21 patients with major depressive disorder and 20 healthy control (HC) subjects. Structural magnetic resonance image data were obtained using a 7-T scanner. Two researchers blinded to the clinical data manually delineated the habenular nuclei and Hb volume, and T1 values were calculated based on overlapping voxels. They found the MDD patients have a smaller right Hb than HC, and that the volume of the right Hb was significantly smaller than that of the left Hb in the MDD group. Generally speaking, this is a well conducted study through my main concern for it the small sample size which may deter the robustness of the results.

1. Introduction: A stronger rationale for the clinical significance of volume and T1 value is needed.

2. Methods: Please provide more information about the MDD patients, such as illness duration.

3. Methods: How to guarantee the accuracy of manually delineation of the habenula?

4. Methods: Details are missing regarding how to calculate the T1 values for each subject.

5. Methods: Did the authors control the quality of MRI data to minimize the head motion?

6. Methods: Did the authors test the normality for those continuous variables, such as age, Hb volume? We recommend plotting the mean and SD if you analyze with an unpaired t test, and the median and interquartile range if you use the nonparametric Mann-Whitney test.

7. Methods: The authors can perform correlation analyses between clinical scale scores and Hb volume and T1 values to link the MRI data to clinical practice.

8. Results: I would suggest adding an iconic image of Hb rather than only axial images to provide a more comprehensive impression of this structure for audiences.

9. Results: I would suggest adding caps to indicate the significant group difference in Figure 3.

10. Discussion: The authors did not discuss how these findings would impact an individual MD patient. In other words, please specify the clinical applications/value of the results.

11. Discussion: More space needs to be devoted to discussing the limitations of this study. For instance, the small sample size which may deter the robustness of the results.

Reviewer #2: This is a paper testing hebanula volume in MDD and laterality of changes in both patients and controls. The paper focuses on n important brain structure, has a clear hypothesis and is well written.

I have some comments:

Introduction:

Lateral and medial hebanula is mentioned. I imagine that the size of the hebanula is too small to allow measurements of these parts separately?

I am not sure if it is right to call hebanula a 'key region', we don't know yet well how crucial it is for MDD development. Perhaps 'important' would do it justice?

One of the exclusion criteria is 'no unstable or major medical or neurological disorders within the past 1 year'. Does it mean that if these were present before, e.g. someone had a stroke, they would be included?

Methods:

Why were two different kinds of a structural image acquired?

A description of a protocol for volume calculation is completely missing and needs to be added (which software was used, which steps were undertaken).

Results:

The MDD group needs to be better characterised. It is important to include the time since MDD onset, possibly the number of episodes, length of the current episode, whether any patient was treatment-resistent. It would be interesting to perform exploratory correlations on the length of disorder and hebanula volume.

It is very important to know whether patients were treated with medications and if so, which medications, as medications may have an impact on both function and structure of the brain regions. The impact of medications should be noted.

In my opinion, the fragment starting at page 11, line 12 ('In the data obtained from...') up to page 13, line 3 belongs to Methods rather that results.

The Authors mention results for T1 but for an average reader it may be unclear what T1 means and why, of all the values, this one was presented as relevant to the subject. A sentence of explanation is needed.

p.13 line 6 - the (mean?) overlapping volumes.

It is worth mentioning that there are scatterplots showing reasults for individual patients and healthy controls.

Discussion

p. 17 line 18 The sentece starting with 'Laterality difference...' is not clear. Does it mean that laterality is necessary to preserve good somatosensory processing in MDD? Or the opposite?

The Authors mention advantages of this study but not limitations. Could you please add a paragraph critically assessing limitations of the study?

6. PLOS authors have the option to publish the peer review history of their article (what does this mean?). If published, this will include your full peer review and any attached files.

---

## [Author Response · Author response to Decision Letter 0]

31 Dec 2020

Responses to Reviewer #1: 

We thank the reviewer for the comments on our article. We believe that the article has been considerably improved based on the reviewer’s comments and suggestions. Please see below the individual responses to their comments.

Comments to the Author: 

In this manuscript, the authors examined the differences in volume and T1 value of the habenula (Hb) between 21 patients with major depressive disorder and 20 healthy control (HC) subjects. Structural magnetic resonance image data were obtained using a 7-T scanner. Two researchers blinded to the clinical data manually delineated the habenular nuclei and Hb volume, and T1 values were calculated based on overlapping voxels. They found the MDD patients have a smaller right Hb than HC, and that the volume of the right Hb was significantly smaller than that of the left Hb in the MDD group. Generally speaking, this is a well conducted study through my main concern for it the small sample size which may deter the robustness of the results.

Comment #1. 

Introduction: A stronger rationale for the clinical significance of volume and T1 value is needed.

Response: Thank you for your comment. As per your suggestion, we have revised the manuscript accordingly. The revised Introduction now reads as follows (Introduction section, page 5, lines 78–85): “Unmedicated patients with bipolar depression showed significantly smaller Hb volumes than an HC group, and currently depressed females with MDD had smaller Hb volumes than healthy females [22]. In another study, hemispheric Hb volumes did not differ between medicated and unmedicated MDD patients and HCs; however, there were significant positive correlations between Hb volumes and depression severity [21]. A recent analysis showed that women with first-episode MDD had higher overall Hb white matter volumes than did healthy controls and patients with treatment-resistant/chronic MDD [20].”

Additionally, (Introduction section, page 6, line 95 – page 7, line 115): “In addition, a new research topic with clinical significance besides asymmetry in the Hb is myelin and T1 relaxation time, which is a measure of how quickly the net magnetization vector recovers to its ground state. After the radiofrequency (RF) pulse is stopped, the absorbed energy is released to the surrounding tissue, and the protons are rearranged in the direction of the external magnetic field in equilibrium [26, 27]. T1 relaxation time is known to be sensitive to the water content in tissues and the macromolecular tissue environment. Changes in water composition can lead to obvious changes in the T1 relaxation time, and measurement of the T1 relaxation time provides a quantitative indicator of the physiological state of the underlying tissue [28, 29]. These T1 relaxation constants have been widely used to study high-intensity lesions such as multiple sclerosis and are used as markers of tissue characteristics in MR imaging studies [30]. A previous study using the T1 relaxation time was also conducted in patients with depression, and unipolar depression patients showed a decreased T1 in the hippocampus compared to healthy controls [31]. In that study, the researchers suggested that the biophysical tissue change might be due to depression [31]. In addition, quantitative T1 mapping has shown the possibility of elucidating cortical myelin content [32], especially in high-resolution 7-T MRI [33]. Strotmann et al. concluded that the low T1 value of the Hb was due to the unusually large number of myelinated fibers [34]. It is likely that decreases in R1 (R1 = 1 / T1 value) are attributable to decreases in myelin in depression [35]. In this context, investigators have begun to use T1 values and R1 as an in vivo assay of myelin content in the Hb [35, 36].”

Comment #2. 

Methods: Please provide more information about the MDD patients, such as illness duration.

Response: As per your suggestion, we have described the clinical information of patients with MDD in the revised manuscript and Table 1 as follows (page 14, line 259 - page 15, line 277): “In addition, the BHS, CGI, and SSI scores were also higher in patients with MDD than in HCs (P < 0.001, see Table 1). Compared to the control group, the depressed group also reported more hopelessness and suicidal ideation, and the overall clinical impression was more severe. When the severity of the MDD patients was classified according to the DSM-5, the number of subjects classified as severe, moderate, and mild severity in the MDD group was 6 (19%), 18 (54%), and 9 (27%), respectively. In the MDD group, the average duration of the current episode was 73.93 weeks, and the average disease duration was 63.45 months. Eighty-two percent of the MDD group were taking antidepressants, and the average duration of antidepressant use was 82.18 weeks (Table 1).”

Comment #3. 

Methods: How to guarantee the accuracy of manually delineation of the habenula?

Response: As mentioned in the manuscript, two examiners manually drew the borders of the habenula and their overlapping ratio of the manually segmented whole Hb was calculated to ensure that the overlap was more than 70%. Then, as suggested by a previous study [44], the area was confined to the overlapped areas, so as not to overestimate the habenular volume. The manuscript now reads as follows (page 10, line 177 – page 11, line 196): “Two well-trained researchers performed the manual delineation of the right and left Hbs using the 7-T MR images (i.e., T1 map) from the subjects. At first, the left and the right Hb were identified based on the selected slices in each data. They were manually traced from the T1 map image according to the following procedure. Using MRIcron and ImageJ as analytic tools, the examiners were able to see the image in all three planes (sagittal, coronal, and axial) simultaneously and to segment manually the target voxels outlining the Hb surface where the signal intensity differs clearly from that of the adjacent brain tissues. These data were used to evaluate the reliability of the Hb definition. During the trace, extra care was taken to separate the Hb boundaries from adjacent, non-Hb brain tissues, such as white matter and cerebrospinal fluid (CSF). This was done as precisely as possible, as confirmed by other senior researchers who had the sufficient research experience in brain MRI. The reliability was determined using the overlap index ratio (%) suggested by a previous study [44]. The overlapping ratio of the manually segmented whole Hb was calculated to ensure that it was more than 70%. Then, the area was confined to the overlapped areas, so as not to overestimate the habenular volume [44].

Overlap Index Ratio (%) = (A ∩ B) / (A ∪ B) × 100 

where A is the number of voxels selected by Examiner #1 and B is the number selected by Examiner #2. The number of voxels (#) within overlapping areas was counted, and then the volume was calculated with the following formula: # � voxel size, where is 0.65�0.65�0.65 (imaging resolution). MRIron (https://www.nitrc.org/projects/mricron) and MATLAB (The Mathworks, Inc.) software were used for segmentation and volume calculations, respectively.”

Comment #4. 

Methods: Details are missing regarding how to calculate the T1 values for each subject.

Response: As per your suggestion, we have inserted the information regarding the estimation of T1 values for each subject in this revised manuscript. It now reads as follows (Methods section, page 12, lines 222–224): “Each voxel has a T1 value estimated from the acquired MRI images, which was introduced in previous studies [43, 45]. The final T1 value for each subject was the average of the T1 values within the overlapped area.” 

Comment #5. 

Methods: Did the authors control the quality of MRI data to minimize the head motion?

Response: As per your suggestion, we have added information on how we minimized head motion in this revised manuscript. It now reads as follows (Methods section, page 9, lines 152-155): “The subjects were asked not to move their heads while they were kept in a comfortable lying position to minimize movement during the scan. In addition, cushions were placed between the RF coil and the subject’s head to secure the head.”

Comment #6. 

Methods: Did the authors test the normality for those continuous variables, such as age, Hb volume? We recommend plotting the mean and SD if you analyze with an unpaired t test, and the median and interquartile range if you use the nonparametric Mann-Whitney test.

Response: To increase the sample size, we performed manual segmentation of the habenula on additional participants. Therefore, the number of participants in the MDD and HC groups was 33 and 36, respectively. After normality was confirmed by the Kolmogorov-Smirnov method, the mean habenula volume of the two groups (MDD and control groups) was compared using a Student's t-test, and the difference between left and right habenula volume within each group was analyzed by a paired t-test. When not satisfying the homogeneity of variance using Levene’s test, the difference was tested using Welch’s t-test. We have revised the description of the statistical analysis in the Methods section (page 13, lines 234-243). 

As per your suggestion, we have plotted the mean and SD in Figure 3. The mean and SD values are also shown in Tables 2 and 3.

Comment #7. 

Methods: The authors can perform correlation analyses between clinical scale scores and Hb volume and T1 values to link the MRI data to clinical practice.

Response: As you advised, to investigate the correlation between the clinical scale scores and Hb volume and T1 values in the depressed group, partial correlation analysis controlling for sex and age was performed. There was no significant correlation among the clinical scale scores, including HDRS-17 score, Hb volume, and T1 values. 

In addition, we performed multiple linear regression analysis to determine whether demographic and clinical characteristics affect the habenula volume of the MDD group. In multivariate regression analyses, variables such as age, sex, years of education, the use of antidepressants, duration of antidepressant use, duration of illness, recurrence of depression, HDRS-17 score, SSI score, and severity of depression were set as independent variables with the stepwise method as they were significantly correlated with whole habenula volume as dependent variable.

In the MDD group, whole Hb volume was associated with age (β =-0.371, p =0.011), male sex (β =-0.485, p =0.001), years of education (β =0.299, p =0.036), and duration of antidepressant use (β =0.331, p =0.014; Table 4). 

We described this issue in the Materials and methods, Results, and Discussion sections and show the statistics in Table 4.

Materials and methods section (page 13, lines 244-250): “In addition, we performed multiple linear regression analysis to determine whether demographic and clinical characteristics affect the Hb volume of the MDD group. In multivariate regression analyses, variables such as age, sex, years of education, the use of antidepressants, duration of antidepressant use, duration of illness, illness recurrence, HDRS-17 score, SSI score, and depression severity were set as independent variables with the stepwise method as they were significantly correlated with whole Hb volume. Whole Hb volume was used as a dependent variable in multivariate linear regression analyses.”

Results section (page 17, line 320– page 18, line 329): “The multivariate linear regression analysis showed the final model was suitable (F = 12.147, P < 0.001) with 62.3% explanatory power (adjusted R2 = 0.623). In the MDD group, whole Hb volume was associated with age (P = 0.011), sex (P = 0.001), years of education (P = 0.036), and duration of antidepressant use (P = 0.014). Among these independent variables, it was found that sex (β = −0.485) had the relatively highest influence on Hb volume, followed by age (β = −0.371), duration of antidepressant use (β = 0.331), and years of education (β = 0.299). (Table 4).” 

Discussion section (page 19, lines 336-339 and page 20, line 365 – page 21, line 383)” Moreover, among the depressed group, it was found that the whole Hb volume was smaller for females and those who were older than for males and those who were younger, respectively, and that the Hb volume increased as the years of education or the duration of antidepressant use was longer.”

“As with previous studies by Savitz et al. [22], it was shown that depressed women had a smaller Hb volume. Another study reported that the Hb volume was smaller when the depression was recurrent or chronic in depressed women [20]. Compared to men in the MDD group, women are affected by female sex hormones, are more sensitive to socio-cultural stressors, and have limited methods of relieving stress [49]; therefore, they might be more vulnerable to Hb volume reduction. In addition, the possibility of sex differences in Hb volume irrespective of depression cannot be excluded [50]. Unlike previous studies that did not find a correlation of Hb volume and age [20, 21], this study showed that the older the individuals in the depressed group, the smaller the volume. The older the patients with MDD, the more likely they are to live in isolation, have less interpersonal contact, and have a higher likelihood of physical illness [51], which may lead to a smaller whole Hb volume. To our knowledge, there are no research results that clearly demonstrate a correlation between Hb volume and duration of education or antidepressant medication use. Past studies suggest that use of antidepressants might have an effect on preventing neuronal damage and cell loss that occur in patients who are depressed [52]. In this context, the longer period of antidepressant use in patients with MDD might be related to a lower decrease in Hb volume. Since pubertal hormones are released and development of higher-order cognition through enhanced plasticity of cortical circuits is present in adolescents [53], the duration of education could affect the volume of the brain. Thus, the longer the education period, the greater the Hb volume. ”

Comment #8. 

Results: I would suggest adding an iconic image of Hb rather than only axial images to provide a more comprehensive impression of this structure for audiences.

Response: As per your suggestion, the additional coronal images are included in Figure 1.

Comment #9. 

Results: I would suggest adding caps to indicate the significant group difference in Figure 3.

Response: As per your advice, we added red caps and marked the significant group difference combinations with an asterisk(*) in Figure 3.

Comment #10. 

Discussion: The authors did not discuss how these findings would impact an individual MD patient. In other words, please specify the clinical applications/value of the results.

Response: Thank you for your comment. Our results, obtained using high-resolution 7-T MRI, could contribute to a better understanding of the neuroanatomical characteristics of Hb in MDD. Although the results of this study are thought to contribute to clarifying the etiology of MDD patients in the future, it is not likely that brain imaging studies can be applied to biomarker for the diagnosis, treatment response, and recurrence of MDD yet. We have described the impact of our findings in the Discussion as follows (page 23, lines 422-427): “This study showed decreased right Hb volume and left-right asymmetry of the Hb volume and T1 value in MDD. Our results, obtained using high-resolution 7-T MRI, could contribute to a better understanding of the neuroanatomical characteristics of Hb in MDD. Although this study is likely to contribute to clarifying the etiology of MDD patients in the future, the brain imaging studies may not be immediately utilized as a biomarker for the diagnosis, treatment response, and recurrence of MDD [66].”

Comment #11. 

Discussion: More space needs to be devoted to discussing the limitations of this study. For instance, the small sample size which may deter the robustness of the results.

Response: Thank you for your comment. Our study included a total of 69 subjects, which could still be considered relatively small. Therefore, we described the potential limitations of our study in the Discussion as per your advice. The contents are as follows (page 23, lines 427–431): “Owing to the relatively small number of subjects in this study, it is difficult to generalize our results. Additionally, the type, amount, and duration of psychotropic medications being taken by each subject in the depression group might have affected the outcome. Further replication studies with more control over such variables and larger sample sizes are needed.”

Responses to reviewer #2 

We thank the reviewer for the comments on our manuscript, which we believe have helped us considerably to improve it. Please see below for individual responses to the reviewer’s comments.

Comments to the Author: 

This is a paper testing hebanula volume in MDD and laterality of changes in both patients and controls. The paper focuses on n important brain structure, has a clear hypothesis and is well written.

Comment #1. 

Introduction:

Lateral and medial hebanula is mentioned. I imagine that the size of the hebanula is too small to allow measurements of these parts separately?

Response: We agree with you. The previous postmortem study divided the habenula into lateral and medial parts; however, it was impossible to distinguish lateral and medial parts using the current 7 Tesla brain imaging technology. Previously, no MRI studies divided the habenula into medial and lateral parts, and the description of lateral and medial habenula in our manuscript only applies to the postmortem study.

Comment #2. 

I am not sure if it is right to call hebanula a 'key region', we don't know yet well how crucial it is for MDD development. Perhaps 'important' would do it justice?

Response: As suggested, we changed “key” to “important” as follows (page 3, line 29 and page 5, line 70): “The habenula (Hb) has been hypothesized to play an essential role in major depressive disorder (MDD) as it is considered to be an important node between fronto-limbic areas and midbrain monoaminergic structures based on animal studies.”

“However, the human Hb, which is estimated to be an important brain region involved in depression, is 5–9 mm in size and neuroimaging studies are lacking because the visualization and exact delineation of this structure is not easy using conventional 3-T brain magnetic resonance imaging (MRI).”

Comment #3. 

One of the exclusion criteria is 'no unstable or major medical or neurological disorders within the past 1 year'. Does it mean that if these were present before, e.g. someone had a stroke, they would be included?

Response: Thank you for this question. Those who had a previous cerebrovascular attack were excluded. To describe this more accurately, we have added the following criteria in the exclusion of the Materials and methods section as follows (page 8, line 134): “(iv) no history of cerebrovascular accident”

Comment #4. 

Methods: Why were two different kinds of a structural image acquired?

Response: MPRAGE images have been the most commonly used images to differentiate the anatomical structures of the brain. However, it was difficult to distinguish small and low-contrast structures such as the habenular using MRPAGE. Therefore, images were acquired using MP2RAGE, a new imaging technique that has the advantage of obtaining T1 maps because multiple images can be acquired simultaneously. However, MPRAGE images are still needed to compare the superiority of MR2RAGE. The revised text now reads as follows (Methods section, page 9, lines 157-162): “Although magnetization-prepared rapid gradient echo (MRPAGE) is commonly used for anatomical imaging, MP2RAGE, a new imaging technique that has the advantage of obtaining T1 relaxation time maps, was used. However, it was not expected to distinguish small and low-contrast structures such as the Hb. Therefore, it was necessary to acquire both images using MPRAGE and MP2RAGE imaging techniques for comparison.”

Comment #5. 

A description of a protocol for volume calculation is completely missing and needs to be added (which software was used, which steps were undertaken).

Response: We inserted information regarding the volume calculation in the revised manuscript (page 11, lines 193-196): “The number of voxels (#) within overlapping areas was counted, and then the volume was calculated with the following formula: # x voxel size, where is 0.65x0.65x0.65 (imaging resolution). MRIron (https://www.nitrc.org/projects/mricron) and MATLAB (The Mathworks, Inc.) software were used for segmentation and volume calculations, respectively.”

Comment #6. 

Results:

The MDD group needs to be better characterised. It is important to include the time since MDD onset, possibly the number of episodes, length of the current episode, whether any patient was treatment-resistent. It would be interesting to perform exploratory correlations on the length of disorder and hebanula volume.

Response: As per your recommendation, we have described the characteristics of the participants in more detail in the revised manuscript as follows (page 14, lines 259-267, and Table 1): “In addition, the BHS, CGI, and SSI scores were also higher in patients with MDD than in HCs (P < 0.001, see Table 1). Compared to the control group, the depressed group also reported more hopelessness and suicidal ideation, and the overall clinical impression was more severe. When the severity of the MDD patients was classified according to the DSM-5, the number of subjects classified as severe, moderate, and mild severity in the MDD group was 6 (19%), 18 (54%), and 9 (27%), respectively. In the MDD group, the average duration of the current episode was 73.93 weeks, and the average disease duration was 63.45 months. Eighty-two percent of the MDD group were taking antidepressants, and the average duration of antidepressant use was 82.18 weeks (Table 1).”

In addition, as you advised, an exploratory partial correlation analysis controlling for sex and age was conducted between clinical characteristics and Hb volume and T1 values in the depressed group. There was no signiﬁcant correlation between the duration of the current episode and Hb volume. 

Comment #7. 

It is very important to know whether patients were treated with medications and if so, which medications, as medications may have an impact on both function and structure of the brain regions. The impact of medications should be noted.

Response: As you pointed out, there is a possibility that the medication could affect the brain volume and function. In the MDD group, 27 participants were taking antidepressants. The effects of clinical information, including medication usage, on the habenula volume were assessed using multiple stepwise regression analysis after performing correlation analysis. A p-value of less than 0.05 (p < 0.05) was considered significant.

We described this issue in the Materials and methods, Results, and Discussion sections and show the statistics in Table 4.

Materials and methods section (page 13, lines 244-250): “In addition, we performed multiple linear regression analysis to determine whether demographic and clinical characteristics affect the Hb volume of the MDD group. In multivariate regression analyses, variables such as age, sex, years of education, the use of antidepressants, duration of antidepressant use, duration of illness, illness recurrence, HDRS-17 score, SSI score, and depression severity were set as independent variables with the stepwise method as they were significantly correlated with whole Hb volume. Whole Hb volume was used as a dependent variable in multivariate linear regression analyses.”

Results section (page 17, line 320 – page 18, line 329): “The multivariate linear regression analysis showed the final model was suitable (F = 12.147, P < 0.001) with 62.3% explanatory power (adjusted R2 = 0.623). In the MDD group, whole Hb volume was associated with age (P = 0.011), sex (P = 0.001), years of education (P = 0.036), and duration of antidepressant use (P = 0.014). Among these independent variables, it was found that sex (β = −0.485) had the relatively highest influence on Hb volume, followed by age (β = −0.371), duration of antidepressant use (β = 0.331), and years of education (β = 0.299). (Table 4)."

Discussion section (page 19, lines 336-339 and page 20, line 365 – page 21, line 383) “Moreover, among the depressed group, it was found that the whole Hb volume was smaller for females and those who were older than for males and those who were younger, respectively, and that the Hb volume increased as the years of education or the duration of antidepressant use was longer.”

“As with previous studies by Savitz et al. [22], it was shown that depressed women had a smaller Hb volume. Another study reported that the Hb volume was smaller when the depression was recurrent or chronic in depressed women [20]. Compared to men in the MDD group, women are affected by female sex hormones, are more sensitive to socio-cultural stressors, and have limited methods of relieving stress [49]; therefore, they might be more vulnerable to Hb volume reduction. In addition, the possibility of sex differences in Hb volume irrespective of depression cannot be excluded [50]. Unlike previous studies that did not find a correlation of Hb volume and age [20, 21], this study showed that the older the individuals in the depressed group, the smaller the volume. The older the patients with MDD, the more likely they areer; SD, Standard Deviation; SSI, Scale for Suicide Ideation. to live in isolation, have less interpersonal contact, and have a higher likelihood of physical illness [51], which may lead to a smaller whole Hb volume. To our knowledge, there are no research results that clearly demonstrate a correlation between Hb volume and duration of education or antidepressant medication use. Past studies suggest that use of antidepressants might have an effect on preventing neuronal damage and cell loss that occur in patients who are depressed [52]. In this context, the longer period of antidepressant use in patients with MDD might be related to a lower decrease in Hb volume. Since pubertal hormones are released and development of higher-order cognition through enhanced plasticity of cortical circuits is present in adolescents [53], the duration of education could affect the volume of the brain. Thus, the longer the education period, the greater the Hb volume.”

Comment #8. 

In my opinion, the fragment starting at page 11, line 12 ('In the data obtained from...') up to page 13, line 3 belongs to Methods rather that results.

Response: We agree with your comment. We moved the parts from the Results to the Methods section (page 11, line 197– page 12, line 231) as per your advice. Thank you for your comment.

Comment #9. 

The Authors mention results for T1 but for an average reader it may be unclear what T1 means and why, of all the values, this one was presented as relevant to the subject. A sentence of explanation is needed.

Response: As per your suggestion, we added an explanation for the T1. It now reads as follows (page 6, line 95 – page 7, line 115): “In addition, a new research topic with clinical significance besides asymmetry in the Hb is myelin and T1 relaxation time, which is a measure of how quickly the net magnetization vector recovers to its ground state. After the radiofrequency (RF) pulse is stopped, the absorbed energy is released to the surrounding tissue, and the protons are rearranged in the direction of the external magnetic field in equilibrium [26, 27]. T1 relaxation time is known to be sensitive to the water content in tissues and the macromolecular tissue environment. Changes in water composition can lead to obvious changes in the T1 relaxation time, and measurement of the T1 relaxation time provides a quantitative indicator of the physiological state of the underlying tissue [28, 29]. These T1 relaxation constants have been widely used to study high-intensity lesions such as multiple sclerosis and are used as markers of tissue characteristics in MR imaging studies [30]. A previous study using the T1 relaxation time was also conducted in patients with depression, and unipolar depression patients showed a decreased T1 in the hippocampus compared to healthy controls [31]. In that study, the researchers suggested that the biophysical tissue change might be due to depression [31]. In addition, quantitative T1 mapping has shown the possibility of elucidating cortical myelin content [32], especially in high-resolution 7-T MRI [33]. Strotmann et al. concluded that the low T1 value of the Hb was due to the unusually large number of myelinated fibers [34]. It is likely that decreases in R1 (R1 = 1 / T1 value) are attributable to decreases in myelin in depression [35]. In this context, investigators have begun to use T1 values and R1 as an in vivo assay of myelin content in the Hb [35, 36].”

Comment #10. 

p.13 line 6 - the (mean?) overlapping volumes.

Response: Yes, it is the mean volume. We have edited this accordingly (page 15, line 280).

Comment #11. 

It is worth mentioning that there are scatterplots showing results for individual patients and healthy controls.

Response: As per your suggestion, we have changed Figure 3 to the scatter plot, including the mean and standard deviation, and we drew red lines between the combinations that differed significantly and marked them with an asterisk(*).

As suggested, we have added an explanation of the scatter plot in the Results section as follows (page 15, line 285 – page 16, line 286): “In Figure 3, the volume and T1 values of the right and left Hb in each group are visualized as scatter plots including the mean and SD.”

Comment #12. 

Discussion

p. 17 line 18 The sentence starting with 'Laterality difference...' is not clear. Does it mean that laterality is necessary to preserve good somatosensory processing in MDD? Or the opposite?

Response: We apologize for the confusion. As per your advice, we have modified the sentences to convey the meaning more clearly as follows (page 21, line 398 – page 22, line 401): “Laterality differences might be necessary to preserve efficient somatosensory processing in MDD [58] and various neurological conditions are associated with loss in the lateralization of brain activity [59-61]; however, it is unclear whether these abnormalities are a cause or consequence of the disease.”

Comment #13. 

The Authors mention advantages of this study but not limitations. Could you please add a paragraph critically assessing limitations of the study?

Response: Thank you for your comment. As per your advice, we have described the limitations of our study in the Discussion section as follows (page 23, lines 424-431): “Although this study is likely to contribute to clarifying the etiology of MDD patients in the future, the brain imaging studies may not be immediately utilized as a biomarker for the diagnosis, treatment response, and recurrence of MDD [66]. Owing to the relatively small number of subjects in this study, it is difficult to generalize our results. Additionally, the type, amount, and duration of psychotropic medications being taken by each subject in the depression group might have affected the outcome. Further replication studies with more control over such variables and larger sample sizes are needed.”

Reference

44. J. Pantel DSOL, K. Cretsinger, H. J. Bockholt, H. Keefe, V. A. Magnotta, and N. C. Andreasen. A new method for the in vivo volumetric measurement of the human hippocampus with high neuroanatomical accuracy. Hippocampus. 2000;10(6).

---

## [Decision Letter · Decision Letter 1]

4 Mar 2021

PONE-D-20-18962R1

Left-right asymmetric and smaller right habenula volume in major depressive disorder on high-resolution 7-T magnetic resonance imaging

PLOS ONE

Dear Dr. kang,

Thank you for submitting your manuscript to PLOS ONE. After careful consideration, we feel that it has merit but does not fully meet PLOS ONE’s publication criteria as it currently stands. Therefore, we invite you to submit a revised version of the manuscript that addresses the points raised during the review process.

We look forward to receiving your revised manuscript.

Kind regards,

Danilo Arnone

Academic Editor

PLOS ONE

Reviewers' comments:

Reviewer's Responses to Questions

**Comments to the Author**

1. If the authors have adequately addressed your comments raised in a previous round of review and you feel that this manuscript is now acceptable for publication, you may indicate that here to bypass the “Comments to the Author” section, enter your conflict of interest statement in the “Confidential to Editor” section, and submit your "Accept" recommendation.

Reviewer #2: All comments have been addressed

Reviewer #3: (No Response)

2. Is the manuscript technically sound, and do the data support the conclusions?

Reviewer #2: Yes

Reviewer #3: Partly

3. Has the statistical analysis been performed appropriately and rigorously? 

Reviewer #2: Yes

Reviewer #3: No

4. Have the authors made all data underlying the findings in their manuscript fully available?

Reviewer #2: Yes

Reviewer #3: No

5. Is the manuscript presented in an intelligible fashion and written in standard English?

Reviewer #2: Yes

Reviewer #3: Yes

6. Review Comments to the Author

Reviewer #2: The Authors have addressed my comments in an adequate manner. The manuscript reads well and the methodology and results are presented in a sound way.

The introduction and discussion have benefitted from information added.

I can recommend the manuscript for publication.

Reviewer #3: Please, find a few comments below for your perusal:

Introduction:

It would be best if you added a reference to the first sentence. It would seem that you are referring to the DSM 5, so please, add the citation. This is not necessarily a universal definition.

Methods:

Please, specify if there was a measure of inter-rater reliability among assessors and if not known, please, add it to the limitations list.

Please, specify if an instrument was used for the diagnostic interview e.g. SCID or MINI and how comorbidity including personality disorders was excluded.

How was mental illness excluded in healthy controls? Did you use a standardized instrument? If nit this is a limitation to be added to the list.

Please, mention all the variables which were extracted from patients and controls irrespective if these were matched or not.

Were the patients experiencing unipolar or bipolar depression?

We would need to know more about the number of episodes and whether you included treatment resistant patients.

Which level of depression did you include? Please, provide the cut off for inclusion according to the rating scale you used. The DSM 5 does not provide a continuous measure of severity. You will need to use the rating scale score for that. This is pretty customary and should be established before inclusion at the stage of study design.

Please, mention which medications patients were taking or if they underwent any other treatment. You would need to specify the time frame between initiation of treatment and time of scanning because medication has most likely an effect on brain structure. Were patients stable on medication and if yes for how long? The above could be added to the discussion as a limitation. See for example:

Arnone, D., McKie, S., Elliott, R. et al. State-dependent changes in hippocampal grey matter in depression. Mol Psychiatry 18, 1265–1272 (2013). https://doi.org/10.1038/mp.2012.150

Please, report the inter-rater reliability for the operators who measured the habenula to show that their measurements were concordant.

Statistical analysis: The approach is unusual. The obvious methodology would be to run an ANCOVA or MANOVA, include left and right structures, healthy controls and depressed subjects in one analysis and control for the confounders within the analysis and not subsequently to check the effect of confounders with a regression model. The regression indicates that confounders had an effect by the way which is concerning in relation to the validity of your t-tests. You would also need to take into account the multiple comparisons (t-tests don’t do that) and the effects of brain volume sex and age (even if matched). Where the patient all right or left handed? If not this is another confounder. In this study particularly, patients had lower level of education vs. controls and were not matched for IQ, it may well be that whole brain morphometry has an impact. Please, run the analysis controlling for potential confounders within the comparisons of the habenula. The analysis you run is not sufficiently valid or strong to support your conclusions.

Results:

Were patients left or right handed? If mixed this needs to be controlled for in the analysis due to laterality effects.

Line 264: The severity is established according to the rating scale and not the DSM 5. This should be established before the study is conducted and is unusual to be mentioned in the results section. Although you could use the number of symptoms for establishing depression severity, you should not use the DSM 5 because it assumes a dichotomous rather than a continuous approach so it really doesn’t provide a measure of severity, the depression rating scale is usually used.

Discussion: You may want to expand on the limitations. I understand that ROI meta-analyses have not reported the habenula in the past because lack of studies. See for example:

D. Arnone, A.M. McIntosh, K.P. Ebmeier, M.R. Munafò, I.M. Anderson. Magnetic resonance imaging studies in unipolar depression: Systematic review and meta-regression analyses, European Neuropsychopharmacology, Volume 22, Issue 1, 2012, Pages 1-16, ISSN 0924-977X, https://doi.org/10.1016/j.euroneuro.2011.05.003. (https://www.sciencedirect.com/science/article/pii/S0924977X11001027)

However, it would seem logical to try to explain why MRI whole brain meta-analyses which used t-maps have not reported abnormalities in this region. Some potential suggestions for discussion depending on whether you included unipolar patients or also bipolar:

Arnone D, Job D, Selvaraj S, Abe O, Amico F, Cheng Y, Colloby SJ, O'Brien JT, Frodl T, Gotlib IH, Ham BJ, Kim MJ, Koolschijn PC, Périco CA, Salvadore G, Thomas AJ, Van Tol MJ, van der Wee NJ, Veltman DJ, Wagner G, McIntosh AM. Computational meta-analysis of statistical parametric maps in major depression. Hum Brain Mapp. 2016 Apr;37(4):1393-404. doi: 10.1002/hbm.23108. Epub 2016 Feb 8. PMID: 26854015; PMCID: PMC6867585.

Stefania Pezzoli, Louise Emsell, Sarah W. Yip, Danai Dima, Panteleimon Giannakopoulos, Mojtaba Zarei, Stefania Tognin, Danilo Arnone, Anthony James, Sven Haller, Sophia Frangou, Guy M. Goodwin, Colm McDonald, Matthew J. Kempton, Meta-analysis of regional white matter volume in bipolar disorder with replication in an independent sample using coordinates, T-maps, and individual MRI data, Neuroscience & Biobehavioral Reviews, Volume 84, 2018, Pages 162-170, ISSN 0149-7634, https://doi.org/10.1016/j.neubiorev.2017.11.005. (https://www.sciencedirect.com/science/article/pii/S0149763417304906)

Wise T, Cleare AJ, Herane A, Young AH, Arnone D. Diagnostic and therapeutic utility of neuroimaging in depression: an overview. Neuropsychiatr Dis Treat. 2014;10:1509-1522. Published 2014 Aug 19. doi:10.2147/NDT.S50156

7. PLOS authors have the option to publish the peer review history of their article (what does this mean?). If published, this will include your full peer review and any attached files.

Reviewer #2: No

Reviewer #3: No

---

## [Author Response · Author response to Decision Letter 1]

11 Apr 2021

Responses to Reviewer #2: 

We deeply appreciate the reviewer’s positive comments on our article.

Responses to Reviewer #3:

We thank the reviewer for their comments regarding our article. We believe that this article has been considerably improved based on the reviewer’s comments and suggestions. Please see below for individual responses to their comments:

Review Comments to the Author: 

Please, find a few comments below for your perusal:

Comment #1. 

Introduction: It would be best if you added a reference to the first sentence. It would seem that you are referring to the DSM 5, so please, add the citation. This is not necessarily a universal definition. 

Response: We agree with your comment. As per your advice, we added the DSM-5 reference to the first sentence of the Introduction section (page 4, line 47; page 28, line 490).

Comment #2. 

Methods: Please, specify if there was a measure of inter-rater reliability among assessors and if not known, please, add it to the limitations list.

Response: The volumes of the segmented structures were utilized for inter-rater reliability (interclass correlation), and the results have been added to the revised manuscript.

 Right p-value Left p-value

ICC (r) 0.703 p<0.001 0.656 p<0.001

r indicates the correlation coefficients.

We have described this issue in the Materials and methods (page 11, lines 203–204) and Results (page 17, lines 314–315) sections.

Material and methods section (page 11, line 203-206): “The volumes of the segmented structures were utilized for inter-rater reliability (interclass correlation [ICC]), and the overlapping ratio of the manually segmented whole Hb was calculated to ensure that it was more than 70%.”

Results section (page 17, line 314–315): “The ICC coefficients were 0.703 (P < 0.001) and 0.656 (P < 0.001) in the right and left Hb, respectively.”

Comment #3. 

Please, specify if an instrument was used for the diagnostic interview e.g. SCID or MINI and how comorbidity including personality disorders was excluded.

Response: Thank you for your comment. As mentioned in the manuscript, board-certified psychiatrists interviewed the participants to assess their eligibility for the study using the Structured Clinical Interview for Diagnostic and Statistical Manual of Mental Disorders, 5th edition (DSM-5) (SCID). In addition, after the participants completed a self-report patient questionnaire for the SCID-5-personality disorder (SCID-5-SPQ) [39], the clinicians reviewed it and asked additional questions to exclude any participants with personality disorder. This information is now included in the Material and methods section (page 8, lines 130–133).

Comment #4. 

How was mental illness excluded in healthy controls? Did you use a standardized instrument? If nit this is a limitation to be added to the list.

Response: As in the above answer, we used the SCID for all participants, and only those who were confirmed not to have mental illness were included in the healthy control group.

Comment #5. 

Please, mention all the variables which were extracted from patients and controls irrespective if these were matched or not.

Response: As per your suggestion, we have described all the variables that were extracted from patients and controls as follows (Methods section, pages 13–14, lines 251–256): “Demographic data and clinical characteristics, including age, sex, education level, and scores of the clinician-administered scales of the HDRS-17, BHS, BDI, CGI, and SSI were summarized and compared using a Student’s t-test or chi-square test. In the MDD group, the number of depressive episodes, duration of current depressive episode, duration of illness, whether taking antidepressants, the duration of antidepressant use, and the subgroups (first episode, treatment-resistance, recurrence, and remission) of MDD were described.”

We also show this clinical information and a comparison between the groups in Table 1.

Comment #6.

Were the patients experiencing unipolar or bipolar depression?

Response: We included only patients with unipolar depression in the MDD group. Patients who had hypomanic or manic episodes were excluded. We described this Method section (page 9, line 144).

Comment #7.

We would need to know more about the number of episodes and whether you included treatment resistant patients.

Response: As suggested, we described the number of depressive episodes and the proportion of treatment-resistant patients as follows (Results section, page 16, lines 291–294, Table 1): “In the MDD group, the average duration of the current episode was 73.93 weeks, the average number of depressive episodes was 2.58 (SD 1.39), the proportion of treatment-resistant patients was 15%, and the average disease duration was 63.45 months.”

We also show this clinical information and a comparison between the groups (Table 1).

Comment #8.

Which level of depression did you include? Please, provide the cut off for inclusion according to the rating scale you used. The DSM 5 does not provide a continuous measure of severity. You will need to use the rating scale score for that. This is pretty customary and should be established before inclusion at the stage of study design.

Response: We included participants who met the DSM-5 criteria for MDD throughout their lifetime in the MDD group, regardless of the severity of the current episode. We rated the severity of their current depression based on the score from the Korean version of the HDRS-17 [41] as follows: within the normal range (0–6), mild depression (7–17), moderate depression (18–24), and severe depression (≥ 25). We also divided the participants into four subgroups according to subgroup criteria described in the manuscript and quoted below.

As per your suggestion, we have described the depression rating scale, severity scoring criteria, and number of participants in each subgroup in the revised manuscript as follows: 

Methods section, page 9, line 143: “The MDD patients who met the diagnostic criteria for MDD as stated in the DSM-5 [40] were included. Patients who had hypomanic or manic episodes were excluded. The MDD group was divided into four subgroups as follows: (i) first episode (those who have experienced the first episode of MDD and have not previously taken psychotropic drugs such as antidepressants, and have a 17-item Hamilton Depression Rating Scale (HDRS-17) score of 7 or higher); (ii) treatment-resistance (those who have experienced major depressive episodes for more than two years in their lifetime and who have sustained major depressive episodes that do not show treatment response to two or more antidepressants); (iii) recurrence (those with two or more episodes of major depressive episodes and have not taken psychiatric medications such as antidepressants for more than one month); and (iv) remission (those with an HDRS-17 score ≤ 6 over the last 2 months or more).”

Methods section, page 9, line 161: “As per a previous study [41], severity on the Korean version of the HDRS-17 was defined based on the total score as follows: within the normal range (0–6), mild depression (7–17), moderate depression (18–24), and severe depression (≥ 25).”

Result section, page 16, line 294: “The MDD group was divided into four subgroups: first episode (n = 12), treatment-resistance (n = 6), recurrence (n = 15), and remission (n = 2). Since some patients belonged to more than one group, the sum of the number of participants of each group exceeded the total number of patients.”

Comment #9.

Please, mention which medications patients were taking or if they underwent any other treatment. You would need to specify the time frame between initiation of treatment and time of scanning because medication has most likely an effect on brain structure. Were patients stable on medication and if yes for how long? The above could be added to the discussion as a limitation. See for example:

Arnone, D., McKie, S., Elliott, R. et al. State-dependent changes in hippocampal grey matter in depression. Mol Psychiatry 18, 1265–1272 (2013). https://doi.org/10.1038/mp.2012.150

Response: As per your suggestion, we have described the antidepressants that the patients had been taking and the period of time between initiation of antidepressant therapy and time of scanning.

It now reads as follows (page 16, lines 297–301): “Eighty-two percent of the MDD group were taking antidepressants, and the average duration of antidepressant use was 82.18 weeks (Table 1). The main antidepressants in patients with MDD were escitalopram (n = 12), vortioxetine (n = 3), bupropion (n = 2), desvenlafaxine (n = 2), fluoxetine (n = 2), sertraline (n = 2), milnacipran (n = 2), and mirtazapine (n = 2).”

As you pointed out, there is a possibility that medication could affect brain function and volume. Therefore, the effects of clinical characteristics, including medication usage, on the habenula volume were assessed using multiple stepwise regression analysis after performing correlation analysis. A P value < 0.05 was considered significant. We described this issue in the Materials and methods, Results, and Discussion sections and show the statistics in Table 4. We also cited the article (reference #55, Arnone et al., 2013) you suggested in the Discussion section as supporting evidence that antidepressants might affect brain structure.

Materials and methods section (page 14, lines 269–275): “In addition, we performed multiple linear regression analysis to determine whether demographic and clinical characteristics affect the Hb volume of the MDD group. In multivariate regression analyses, variables such as age, sex, years of education, the use of antidepressants, duration of antidepressant use, duration of illness, illness recurrence, HDRS-17 score, SSI score, and depression severity were set as independent variables with the stepwise method as they were significantly correlated with whole Hb volume. Whole Hb volume was used as a dependent variable in multivariate linear regression analyses.”

Results section (page 20, line 361–367): “The multivariate linear regression analysis showed the final model was suitable (F = 12.147, P < 0.001) with 62.3% explanatory power (adjusted R2 = 0.623). In the MDD group, whole Hb volume was associated with age (P = 0.011), sex (P = 0.001), years of education (P = 0.036), and duration of antidepressant use (P = 0.014). Among these independent variables, it was found that sex (β = −0.485) had the relatively highest influence on Hb volume, followed by age (β = −0.371), duration of antidepressant use (β = 0.331), and years of education (β = 0.299). (Table 4).”

Discussion section (page 22, lines 380): “Moreover, among the depressed group, it was found that the whole Hb volume was smaller for females and those who were older than for males and those who were younger, respectively, and that the Hb volume increased as the years of education or the duration of antidepressant use was longer.”

Discussion section (page 24, lines 419–427): “To our knowledge, there are no research results that clearly demonstrate a correlation between Hb volume and duration of education or antidepressant medication use. Past studies suggest that use of antidepressants might have an effect on preventing neuronal damage and cell loss that occur in patients who are depressed [54, 55]. In this context, the longer period of antidepressant use in patients with MDD might be related to a lower decrease in Hb volume. Since pubertal hormones are released and development of higher-order cognition through enhanced plasticity of cortical circuits is present in adolescents [56], the duration of education could affect the volume of the brain. Thus, the longer the education period, the greater the Hb volume.”

We also described this issue as a limitation as follows (page 26, lines 483–484): “Additionally, the type, amount, and duration of psychotropic medications being taken by each subject in the depression group might have affected the outcome.”

r indicates the correlation coefficients.

We have described this issue in the Materials and methods (page 11, lines 203–204) and Results (page 17, lines 314–315) sections.

Comment #10.

Please, report the inter-rater reliability for the operators who measured the habenula to show that their measurements were concordant.

Response: As we responded to Comment #2, the volumes of the segmented structures were utilized for inter-rater reliability (interclass correlation), and the results have been added to the revised manuscript.

Comment #11.

Statistical analysis: The approach is unusual. The obvious methodology would be to run an ANCOVA or MANOVA, include left and right structures, healthy controls and depressed subjects in one analysis and control for the confounders within the analysis and not subsequently to check the effect of confounders with a regression model. The regression indicates that confounders had an effect by the way which is concerning in relation to the validity of your t-tests. You would also need to take into account the multiple comparisons (t-tests don’t do that) and the effects of brain volume sex and age (even if matched). Where the patient all right or left handed? If not this is another confounder. In this study particularly, patients had lower level of education vs. controls and were not matched for IQ, it may well be that whole brain morphometry has an impact. Please, run the analysis controlling for potential confounders within the comparisons of the habenula. The analysis you run is not sufficiently valid or strong to support your conclusions. This is consistent with our initial study results, where the difference in the volume of the right habenula significantly differed between the two groups.

Response: As described in the paper, all subjects were right-handed, since the common eligibility criteria included ‘right-handed using the Edinburgh Handedness Test’ (page 8, lines 135–136).

As per your suggestion, we performed an ANCOVA to compare habenula volume while controlling for potential confounding factors (i.e., age, sex, education level, and total intracranial volume). Since individual differences in whole-brain volume could have been a confounder in the analysis, we divided each habenula volume by the whole brain volume and analyzed the volume ratio. The normalization of the Hb volumes was performed using the total intracranial volume (ICV) from the 3T MRI. The Hb volumes were divided by the ICV for each participant ((Hb Volume)/ICV×100) to adjust for individual differences in brain size. According to the results of the ANCOVA, which included age, sex, and education level as covariates, there was a significant difference in the volume ratio of the right Hb between the MDD and healthy control groups (F = 5.544, P = 0.022). This is consistent with our initial study results, in which the volume of the right Hb differed significantly between the two groups. The partial eta-squared values (representing the effect size) for the left and right Hb volume ratios were small (η2=0.001) and relatively large (η2=0.080), respectively. Supplementary Table S1 provides a summary of the ANCOVA results. 

This has been described in the Methods, Results, and Discussion sections as follows:

Methods section (page 14, lines 263–268): “Furthermore, to adjust for potential confounding factors, analysis of covariance (ANCOVA) was conducted comparing the normalized right and left Hb volumes between the MDD and HC groups, controlling for age, sex, education level, and individual differences in brain size. The normalization of the Hb volumes was performed using the total intracranial volume (ICV) from the 3T MRI. The Hb volumes were divided by the ICV for each participant ((Hb Volume)/ICV×100) to adjust for individual differences in brain size.”

Results section (page 19-20, lines 354-360): “According to the results of the ANCOVA, which included age, sex, and education level as covariates, there was a significant difference in the right Hb volume ratio between the MDD and normal groups (F = 5.544, p = 0.022, Supplementary Table S1). This is consistent with the results from the Student’s t-test analysis, in which the volume of the right Hb significantly differed between the two groups. The partial eta-squared values (representing the effect size) for the left and right Hb volume ratios were small (η2 = 0.001) and relatively large (η2 = 0.080), respectively.”

Discussion section (page 22, lines 377-378): “In addition, in the analysis that included age, sex, and education level as covariates, the right Hb volume ratio was significantly different between the healthy and depressed groups.”

Comment #12.

Results: Were patients left or right handed? If mixed this needs to be controlled for in the analysis due to laterality effects.

Response: As described in the manuscript, we performed the Edinburgh Handedness Test and included only subjects identified as right-handed, as described in the eligibility criteria.

Handedness is described as follows (page 8, lines 135–136): “The common eligibility criteria are follows: (i) aged 20–65 years; (ii) identified as right-handed using the Edinburgh Handedness Test;”

Comment #12.

Line 264: The severity is established according to the rating scale and not the DSM 5. This should be established before the study is conducted and is unusual to be mentioned in the results section. Although you could use the number of symptoms for establishing depression severity, you should not use the DSM 5 because it assumes a dichotomous rather than a continuous approach so it really doesn’t provide a measure of severity, the depression rating scale is usually used.

Response: As per your advice, we have added the following in the Methods and Results sections and to Table 1.

Methods section (page 9, lines 161-163): “As per a previous study [41], severity on the Korean version of the HDRS-17 was defined based on the total score as follows: within the normal range (0–6), mild depression (7–17), moderate depression (18–24), and severe depression (≥ 25).”

Results sections (page 16, lines 289–291): “When the severity of depression was classified according to the range of the total score on the HDRS-17, 38 (55%), 14 (20%), 13 (19%), and 4 (6%) were classified as mild, moderate, and severe depression, respectively.” 

Comment #13.

Discussion: You may want to expand on the limitations. I understand that ROI meta-analyses have not reported the habenula in the past because lack of studies. See for example:

D. Arnone, A.M. McIntosh, K.P. Ebmeier, M.R. Munafò, I.M. Anderson. Magnetic resonance imaging studies in unipolar depression: Systematic review and meta-regression analyses, European Neuropsychopharmacology, Volume 22, Issue 1, 2012, Pages 1-16, ISSN 0924-977X, https://doi.org/10.1016/j.euroneuro.2011.05.003. (https://www.sciencedirect.com/science/article/pii/S0924977X11001027)

However, it would seem logical to try to explain why MRI whole brain meta-analyses which used t-maps have not reported abnormalities in this region. Some potential suggestions for discussion depending on whether you included unipolar patients or also bipolar:

Arnone D, Job D, Selvaraj S, Abe O, Amico F, Cheng Y, Colloby SJ, O'Brien JT, Frodl T, Gotlib IH, Ham BJ, Kim MJ, Koolschijn PC, Périco CA, Salvadore G, Thomas AJ, Van Tol MJ, van der Wee NJ, Veltman DJ, Wagner G, McIntosh AM. Computational meta-analysis of statistical parametric maps in major depression. Hum Brain Mapp. 2016 Apr;37(4):1393-404. doi: 10.1002/hbm.23108. Epub 2016 Feb 8. PMID: 26854015; PMCID: PMC6867585.

Stefania Pezzoli, Louise Emsell, Sarah W. Yip, Danai Dima, Panteleimon Giannakopoulos, Mojtaba Zarei, Stefania Tognin, Danilo Arnone, Anthony James, Sven Haller, Sophia Frangou, Guy M. Goodwin, Colm McDonald, Matthew J. Kempton, Meta-analysis of regional white matter volume in bipolar disorder with replication in an independent sample using coordinates, T-maps, and individual MRI data, Neuroscience & Biobehavioral Reviews, Volume 84, 2018, Pages 162-170, ISSN 0149-7634, https://doi.org/10.1016/j.neubiorev.2017.11.005. (https://www.sciencedirect.com/science/article/pii/S0149763417304906)

Wise T, Cleare AJ, Herane A, Young AH, Arnone D. Diagnostic and therapeutic utility of neuroimaging in depression: an overview. Neuropsychiatr Dis Treat. 2014;10:1509-1522. Published 2014 Aug 19. doi:10.2147/NDT.S50156

Response: Thank you for your comment. As you mentioned, previous ROI meta-analyses did not include individual studies regarding the habenula, so it was impossible to determine an association between the habenula and depression. In addition, there were four differences between this study and previous studies. First, the subjects of one meta-analysis (Pezzoli et al., 2018) were patients with bipolar depression, while our study focused only on patients with unipolar depressive disorder. Second, the other meta-analyses (Aaron et al., 2012 & 2016) differed in the diagnostic system and scales for diagnosing MDD. Our study diagnosed MDD using the DSM-5 diagnostic system, but the papers in the meta-analyses used other diagnostic systems such as the DSM-III & IV, Schedule for Affective Disorders and Schizophrenia (SADS), Kiddie Schedule for Affective Disorders and Schizophrenia, and ICD-10. Third, the previous studies do not seem to have used structured interviews such as the SCID when selecting subjects. Fourth, the habenula is a structure included in both white and gray matter; however, previous studies were conducted only in white matter or gray matter; therefore, it may have been difficult to find an association between depression and habenula. Fifth, the habenula can be visualized in MRIs from 7.0 Tesla scanners due to improved spatial resolution compared to lower tesla scanners, so the resolution may have been insufficient to clearly delineate the habenula in previous studies that used lower tesla MRIs. 

Therefore, we revised the manuscript in the Conclusion section as follows (page 25, lines 452–460): “Previous studies, including meta-analyses, have investigated structural brain abnormalities in depression [65-67]; however, our study is distinct from previous studies. In most previous studies, the Hb were not designated as ROIs, and most studies used lower resolution brain MRIs from 3T, 1.5T, or 1.0T scanners [65, 66]. In addition, the clinical diagnoses of the participants were different (unipolar depression in our study vs. bipolar depression in a previous study [67]), or the diagnostic system or clinical scales [65, 66] used in the studies were dissimilar to ours. Moreover, there were differences in the treatment settings, brain imaging techniques, and methods of obtaining and analyzing the ROIs between the current and previous studies [65-67].”

Given your suggestion, we described the limitations as follows (page 26, lines 477–486): “Although the results of this study are thought to contribute to clarifying the etiology of MDD patients in the future, it is not likely that brain imaging studies can be applied to biomarker for the diagnosis, treatment response, and recurrence of MDD yet [72]. Owing to the relatively small number of subjects in this study, it is difficult to generalize our results. In addition, more advanced neuroimaging analyses and techniques such as machine learning methods should be attempted to further reveal evidence regarding the association between depression and the Hb [73]. Additionally, the type, amount, and duration of psychotropic medications being taken by each subject in the depression group might have affected the outcome. Further replication studies with more control over such variables and larger sample sizes are needed.”

---

## [Decision Letter · Decision Letter 2]

11 May 2021

PONE-D-20-18962R2

Left-right asymmetric and smaller right habenula volume in major depressive disorder on high-resolution 7-T magnetic resonance imaging

PLOS ONE

Dear Dr. kang,

Thank you for submitting your manuscript to PLOS ONE. After careful consideration, we feel that it has merit but does not fully meet PLOS ONE’s publication criteria as it currently stands. Therefore, we invite you to submit a revised version of the manuscript that addresses the points raised during the review process.

We look forward to receiving your revised manuscript.

Kind regards,

Danilo Arnone

Academic Editor

PLOS ONE

Reviewers' comments:

Reviewer's Responses to Questions

**Comments to the Author**

1. If the authors have adequately addressed your comments raised in a previous round of review and you feel that this manuscript is now acceptable for publication, you may indicate that here to bypass the “Comments to the Author” section, enter your conflict of interest statement in the “Confidential to Editor” section, and submit your "Accept" recommendation.

Reviewer #3: (No Response)

2. Is the manuscript technically sound, and do the data support the conclusions?

Reviewer #3: No

3. Has the statistical analysis been performed appropriately and rigorously? 

Reviewer #3: No

4. Have the authors made all data underlying the findings in their manuscript fully available?

Reviewer #3: No

5. Is the manuscript presented in an intelligible fashion and written in standard English?

Reviewer #3: Yes

6. Review Comments to the Author

Reviewer #3: I can see that the authors made some changes. The statistical analysis remains a concern.

The self-report questionnaire for personality disorder is not the same as a clinical interview. Please, discuss in the limitations as patients may still have a personality disorder.

Please, specify in your inclusion/exclusion criteria whether patients met criteria for any disorder other than MDD. It is essential to comment on comorbidities. As you know anxiety disorders (and not only) are highly comorbid with MDD.

How many clinicians assessed the participants? Did you check inter-rater reliability?

Need a citation for the Edinburgh test.

It appears that paired t tests is still used as the main analysis. This is incorrect in my view as this doesn’t consider the confounders. Please report the results as ANCOVA and please, correct the level of significance for the number of regions you are investigating (2 in this case).

The majority of the patients suffered a mild form of illness. This is a major limitation because how representative of MDD is mild illness? The majority of the studies investigate moderate to severe levels of illness. My suggestion is to discuss this in the limitation as it might at least contribute to explain some of your findings.

Did any patient used more than one psychotropic medication? Did anyone use anything else other than antidepressants? The use of psychotropics is a major limitation.

I don’t understand the results. Why are you still referring to paired t tests? This approach is not correct as it does not consider all the confounders and individual brain differences. Please, use ANCOVA throughout and amend table 2 and 3 which are still referring to paired t test. Really confusing.

This study has several limitations. I cannot see that limitations were addressed in the discussion. All the points mentioned here and in the previous round of comments including medication status, severity of illness, comorbidities, etc., could be a starting point.

7. PLOS authors have the option to publish the peer review history of their article (what does this mean?). If published, this will include your full peer review and any attached files.

Reviewer #3: No

---

## [Author Response · Author response to Decision Letter 2]

22 Jun 2021

Responses to Reviewer #3:

We thank the reviewer for their comments regarding our article. We believe that this article has been considerably improved based on the reviewer’s comments and suggestions. Please see below for individual responses to their comments:

Review Comments to the Author

I can see that the authors made some changes. The statistical analysis remains a concern.

Comment #1. 

The self-report questionnaire for personality disorder is not the same as a clinical interview. Please, discuss in the limitations as patients may still have a personality disorder.

Response: We agree with your comment. As per your advice, we described the limitations of using the self-report questionnaire (SCID-5-SPQ) to exclude participants with personality disorders as follows:

Limitations section (page 26, lines 490–493): “Due to using self-reported measures, not structured interviews, to screen for personality disorders, there is a possibility of self-report bias. Therefore, we might not have accurately excluded participants with personality disorders.”

Comment #2. 

Please, specify in your inclusion/exclusion criteria whether patients met criteria for any disorder other than MDD. It is essential to comment on comorbidities. As you know anxiety disorders (and not only) are highly comorbid with MDD.

Response: As per your advice, we have added this in the Methods section (page 9, lines 144-153) as follows: “In addition, participants in the MDD group did not have any of the following psychiatric comorbidities: schizophrenia spectrum and other psychotic disorders (delusional disorder, brief psychotic disorder, schizophreniform disorder, schizophrenia, schizoaffective disorder, catatonia); major anxiety disorders (panic disorder, social anxiety disorder, specific phobia); obsessive-compulsive and related disorders (obsessive-compulsive disorder, body dysmorphic disorder, hoarding disorder, trichotillomania, excoriation disorder); or disruptive, impulse-control, and conduct disorders (oppositional defiant disorder, intermittent explosive disorder, conduct disorder). If a patient's psychiatric symptoms could be explained as being due to MDD, we did not specify an additional diagnosis for the patient according to DSM-5.”

Comment #3. 

How many clinicians assessed the participants? Did you check inter-rater reliability?

Response: Only one clinician (SGK) evaluated all the participants. After the initial prescreening, board-certified psychiatrist (SGK) interviewed all participants to assess their eligibility for the present study using the Structured Clinical Interview for Diagnostic and Statistical Manual of Mental Disorders, 5th edition (SCID). In addition, after the participants completed a self-report patient questionnaire for the SCID-5-personality disorder (SCID-5-SPQ), the clinician reviewed it and asked additional questions to exclude any participants with personality disorder.

Comment #4. 

Need a citation for the Edinburgh test.

Response: Thank you for the comment. As per your suggestion, we added the reference for the Edinburgh Inventory to the first sentence of the second paragraph of the Methods section (page 8, line 136) as follows: “The common eligibility criteria are follows: (i) aged 20–65 years; (ii) identified as right-handed using the Edinburgh Handedness Test [40]; (iii) no unstable or major medical or neurological disorders within the past 1 year; (iv) no history of cerebrovascular accident; (v) no history ono substance use disorder within the past 1 year; (vi) no personality disorder, intellectual disability, or neurocognitive disorders; (vii) no current serious suicide risk; (viii) no history of significant brain injury or previous abnormal findings on brain imaging; (ix) no relative or absolute contraindications for MRI (e.g., metal material in the body); and (x) not pregnant or lactating.”

Reference

40. Oldfield R. The assessment and analysis of handedness: The Edinburgh inventory. Neuropsychologica. 1971;9: 97–113.

Comment #5.

It appears that paired t tests is still used as the main analysis. This is incorrect in my view as this doesn’t consider the confounders. Please report the results as ANCOVA and please, correct the level of significance for the number of regions you are investigating (2 in this case).

Response: We have accommodated your concerns and made the following corrections, as you recommended. For comparisons between the MDD and HC groups, we performed an ANCOVA analysis controlling for confounders rather than conducting a Student's t-test analysis. We also adjusted the significance level to 0.05/2 using the Bonferroni method as you advised, so we only rejected a null hypothesis if the p-value was less than 0.025. We presented the ANCOVA analysis as the main result (Table 2), and the results derived by Student’s t-tests were transferred to the supplementary material. We described this in the Methods and Results sections as follows:

Material and methods section (page 14, line 267–page 15, line 280): “To adjust for potential confounding factors, analysis of covariance (ANCOVA) was conducted comparing the normalized right and left Hb volumes between the MDD and HC groups, controlling for age, sex, education level, and individual differences in brain size. The normalization of the Hb volumes was performed using the total intracranial volume (ICV) from the 3T MRI. The Hb volumes were divided by the ICV for each participant ((Hb Volume)/ICV×100) to adjust for individual differences in brain size. Since the number of regions we investigated in this analysis was two, the Bonferroni adjusted level of significance (alpha) was calculated to be 2.5%. We rejected the null hypothesis when the p-value was less than the adjusted alpha.

Furthermore, the group difference (MDD vs. HC for volume and T1 value of the right and left Hbs) and the inter-hemispheric difference (right vs. left Hb volume and T1 value) in each group were analyzed using Student’s t-test and paired t-test, respectively. Normality was confirmed using the Kolmogorov-Smirnov method. When not satisfying the homogeneity of variance using Levene’s test, the difference was tested using Welch’s t-test.”

Results section (page 19, line 358–page 20, line 366): “In the between group comparison, according to the results of the ANCOVA, which included age, sex, and education level as covariates, there was a significant difference in the right Hb volume ratio between the MDD and HC groups (F = 5.544, p = 0.022, Table 2). The partial eta-squared values (representing the effect size) for the left and right Hb volume ratios were small (η2 = 0.001) and relatively large (η2 = 0.080), respectively. In the analysis using Student’s t-test, participants with MDD had a significantly smaller right Hb volume than those in the HC group (P = 0.018, Supplementary table S1). There was no significant difference in the left Hb volume between the two groups. This is consistent with the results from the ANCOVA, in which the volume of the right Hb differed significantly between the two groups.”

Comment #6.

The majority of the patients suffered a mild form of illness. This is a major limitation because how representative of MDD is mild illness? The majority of the studies investigate moderate to severe levels of illness. My suggestion is to discuss this in the limitation as it might at least contribute to explain some of your findings.

Response: As you mentioned, 42% (n = 14) of patients with MDD had mild depression based on the HDRS score on the MRI scanning day. However, it is presumed that this is because the MDD group had been continuously receiving treatment, such as medication and psychotherapy. In fact, 51% of the patients with MDD had recurrence (n = 15) and remission (n = 2) of major depressive episodes. In general, the state of depression is not directly related to the structural state of the brain. However, as you suggested, since the severity of depressive symptoms was different, this might have affected the outcomes of this study. Therefore, we have described this in the Limitations section It now reads as follows (page 27, lines 500–504): “Additionally, the severity of MDD symptoms and the type, amount, and duration of psychotropic medications being taken by each participant in the MDD group might have affected the outcomes. In future research, it is essential to compare subdivided brain regions according to the patients’ severity of symptoms, whether or not they are taking any medications, and medication type.”

Comment #7.

Did any patient used more than one psychotropic medication? Did anyone use anything else other than antidepressants? The use of psychotropics is a major limitation.

Response: As per your suggestion, we have described the use of psychotropics in participants in the MDD group. We have described this in the Results and Limitations sections as follows:

Result section, page 17, line 323–page 18, line 326: “In the MDD group, other psychotropic drugs were being taken as follows: benzodiazepine (n = 17), zolpidem (n = 2), quetiapine (n = 7), aripiprazole (n = 6), and olanzapine (n = 1). Second-generation antipsychotics were prescribed as adjunctive treatments for MDD.”

Limitations section, page 27, lines 500–505: “Additionally, the severity of MDD symptoms and the type, amount, and duration of psychotropic medications being taken by each participant in the MDD group might have affected the outcomes. In future research, it is essential to compare subdivided brain regions according to the patients’ severity of symptoms, whether or not they are taking any medications, and medication type. Further replication studies with more control over such variables and larger sample sizes are needed.”

”

Comment #8.

I don’t understand the results. Why are you still referring to paired t tests? This approach is not correct as it does not consider all the confounders and individual brain differences. Please, use ANCOVA throughout and amend table 2 and 3 which are still referring to paired t test. Really confusing.

Response: We have accommodated your concerns and made the following corrections, as we also noted in our response to Comment #5. For comparisons between the MDD and HC groups, we performed an ANCOVA analysis controlling for confounders rather than conducting a Student's t-test analysis. We also adjusted the significance level to 0.05/2 using the Bonferroni method as you advised, so we only rejected a null hypothesis if the p-value was less than 0.025. We presented the ANCOVA analysis as the main result (Table 2), and the results derived by Student’s t-tests were transferred to the supplementary material. We described this in the Methods and Results sections as follows:

Material and methods section (page 14, line 267–page 15, line 280): “To adjust for potential confounding factors, analysis of covariance (ANCOVA) was conducted comparing the normalized right and left Hb volumes between the MDD and HC groups, controlling for age, sex, education level, and individual differences in brain size. The normalization of the Hb volumes was performed using the total intracranial volume (ICV) from the 3T MRI. The Hb volumes were divided by the ICV for each participant ((Hb Volume)/ICV×100) to adjust for individual differences in brain size. Since the number of regions we investigated in this analysis was two, the Bonferroni adjusted level of significance (alpha) was calculated to be 2.5%. We rejected the null hypothesis when the p-value was less than the adjusted alpha.

Furthermore, the group difference (MDD vs. HC for volume and T1 value of the right and left Hbs) and the inter-hemispheric difference (right vs. left Hb volume and T1 value) in each group were analyzed using Student’s t-test and paired t-test, respectively. Normality was confirmed using the Kolmogorov-Smirnov method. When not satisfying the homogeneity of variance using Levene’s test, the difference was tested using Welch’s t-test.”

Results section (page 19, line 357–page 20, line 366): “In the between group comparison, according to the results of the ANCOVA, which included age, sex, and education level as covariates, there was a significant difference in the right Hb volume ratio between the MDD and HC groups (F = 5.544, p = 0.022, Table 2). The partial eta-squared values (representing the effect size) for the left and right Hb volume ratios were small (η2 = 0.001) and relatively large (η2 = 0.080), respectively. In the analysis using Student’s t-test, participants with MDD had a significantly smaller right Hb volume than those in the HC group (P = 0.018, Supplementary table S1). There was no significant difference in the left Hb volume between the two groups. This is consistent with the results from the ANCOVA, in which the volume of the right Hb differed significantly between the two groups.”

However, it does not seem appropriate to use ANCOVA when analyzing inter-hemispheric differences within the same group. ANOVA is a method used to compare the mean of the dependent variable between two or more different groups. Therefore, we performed a paired t-test to compare the mean values of the left and right areas within the same group.

Comment #9.

This study has several limitations. I cannot see that limitations were addressed in the discussion. All the points mentioned here and in the previous round of comments including medication status, severity of illness, comorbidities, etc., could be a starting point.

Response: As per your suggestion, we separated the Limitation section from the Discussion section and described these issues as limitations as follows (page 26, line 487–page 27, line 505): 

“Limitations

Although the results of this study are thought to contribute to clarifying the etiology of MDD patients in the future, it is not likely that brain imaging studies can be applied to biomarker for the diagnosis, treatment response, and recurrence of MDD yet [72]. Due to using self-reported measures, not structured interviews, to screen for personality disorders, there is a possibility of self-report bias. Therefore, we might not have accurately excluded participants with personality disorders. In addition, although we attempted to exclude participants with psychiatric disorders other than MDD under the DSM-5, it would have been difficult to entirely exclude individuals who might have also had these other disorders. Thus, there is a possibility that structural changes in the brain were caused by comorbid disorders. Owing to the relatively small number of subjects in this study, it is difficult to generalize our results. In addition, more advanced neuroimaging analyses and techniques such as machine learning methods should be attempted to further reveal evidence regarding the association between depression and the Hb [73]. Additionally, the severity of MDD symptoms and the type, amount, and duration of psychotropic medications being taken by each participant in the MDD group might have affected the outcomes. In future research, it is essential to compare subdivided brain regions according to the patients’ severity of symptoms, whether or not they are taking any medications, and medication type. Further replication studies with more control over such variables and larger sample sizes are needed.”

---

## [Decision Letter · Decision Letter 3]

19 Jul 2021

Left-right asymmetric and smaller right habenula volume in major depressive disorder on high-resolution 7-T magnetic resonance imaging

PONE-D-20-18962R3

Dear Dr. kang,

We’re pleased to inform you that your manuscript has been judged scientifically suitable for publication and will be formally accepted for publication once it meets all outstanding technical requirements.

Kind regards,

Danilo Arnone

Academic Editor

PLOS ONE

Additional Editor Comments (optional):

Please, review data access policy.

Reviewers' comments:

Reviewer's Responses to Questions

**Comments to the Author**

1. If the authors have adequately addressed your comments raised in a previous round of review and you feel that this manuscript is now acceptable for publication, you may indicate that here to bypass the “Comments to the Author” section, enter your conflict of interest statement in the “Confidential to Editor” section, and submit your "Accept" recommendation.

Reviewer #3: All comments have been addressed

2. Is the manuscript technically sound, and do the data support the conclusions?

Reviewer #3: Yes

3. Has the statistical analysis been performed appropriately and rigorously? 

Reviewer #3: Yes

4. Have the authors made all data underlying the findings in their manuscript fully available?

Reviewer #3: No

5. Is the manuscript presented in an intelligible fashion and written in standard English?

Reviewer #3: Yes

6. Review Comments to the Author

Reviewer #3: NO further comments, all comments have been addressed in the revised version of the manuscripts as suggested

7. PLOS authors have the option to publish the peer review history of their article (what does this mean?). If published, this will include your full peer review and any attached files.

Reviewer #3: No

---

## [Editor Report · Acceptance letter]

23 Jul 2021

PONE-D-20-18962R3 

Left-right asymmetric and smaller right habenula volume in major depressive disorder on high-resolution 7-T magnetic resonance imaging 

Dear Dr. Kang:

I'm pleased to inform you that your manuscript has been deemed suitable for publication in PLOS ONE. Congratulations! Your manuscript is now with our production department. 

Kind regards, 

on behalf of

Dr. Danilo Arnone 

Academic Editor

PLOS ONE